# MACF1 controls skeletal muscle function through the microtubule-dependent localization of extra-synaptic myonuclei and mitochondria biogenesis

Alireza Ghasemizadeh[1], Emilie Christin[1], Alexandre Guiraud[1], Nathalie Couturier[1], Marie Abitbol[1,2], Valerie Risson[1], Emmanuelle Girard[1], Christophe Jagla[3], Cedric Soler[3], Lilia Laddada[3], Colline Sanchez[1], Francisco-Ignacio Jaque-Fernandez[1], Vincent Jacquemond[1], Jean-Luc Thomas[1], Marine Lanfranchi[1], Julien Courchet[1], Julien Gondin[1], Laurent Schaeffer[1], Vincent Gache[1]*

[1]Institut NeuroMyoGène, CNRS UMR5310, INSERM U1217, Faculté de Médecine Rockefeller, Université Claude Bernard Lyon I, Lyon Cedex, France; [2]Université Marcy l'Etoile, VetAgro Sup, Lyon, France; [3]GReD Laboratory, Clermont-Auvergne University, INSERM U1103, CNRS, Clermont-Ferrand, France

*For correspondence:
vincent.gache@inserm.fr

Competing interests: The authors declare that no competing interests exist.

**Abstract** Skeletal muscles are composed of hundreds of multinucleated muscle fibers (myofibers) whose myonuclei are regularly positioned all along the myofiber's periphery except the few ones clustered underneath the neuromuscular junction (NMJ) at the synaptic zone. This precise myonuclei organization is altered in different types of muscle disease, including centronuclear myopathies (CNMs). However, the molecular machinery regulating myonuclei position and organization in mature myofibers remains largely unknown. Conversely, it is also unclear how peripheral myonuclei positioning is lost in the related muscle diseases. Here, we describe the microtubule-associated protein, MACF1, as an essential and evolutionary conserved regulator of myonuclei positioning and maintenance, in cultured mammalian myotubes, in *Drosophila* muscle, and in adult mammalian muscle using a conditional muscle-specific knockout mouse model. In vitro, we show that MACF1 controls microtubules dynamics and contributes to microtubule stabilization during myofiber's maturation. In addition, we demonstrate that MACF1 regulates the microtubules density specifically around myonuclei, and, as a consequence, governs myonuclei motion. Our in vivo studies show that MACF1 deficiency is associated with alteration of extra-synaptic myonuclei positioning and microtubules network organization, both preceding NMJ fragmentation. Accordingly, MACF1 deficiency results in reduced muscle excitability and disorganized triads, leaving voltage-activated sarcoplasmic reticulum Ca$^{2+}$ release and maximal muscle force unchanged. Finally, adult MACF1-KO mice present an improved resistance to fatigue correlated with a strong increase in mitochondria biogenesis.

## Introduction

Throughout skeletal muscle development, myonuclei actively move to settle at specific locations in the fully differentiated muscle fibers (myofibers) where they are regularly spaced along the longitudinal fiber axis, sitting at the periphery underneath the surface membrane (*Roman and Gomes, 2018*). The density of myonuclei along myofibers' length remains relatively constant with the exception of the neuromuscular junction (NMJ), where ~5–6 myonuclei (synaptic myonuclei) are clustered immediately below the NMJ (*Bruusgaard et al., 2006*; *Manhart et al., 2018*; *Ravel-Chapuis et al., 2007*).

Failure in proper extra-synaptic myonuclei patterning has been linked to different myopathies and muscle weakness (*Collins et al., 2017*; *Falcone et al., 2014*; *Metzger et al., 2012*; *Perillo and Folker, 2018*; *Roman et al., 2017*). This precise organization is correlated with a particular 'flatten' shape of the myonuclei, whose alteration has recently emerged as a potential contributor to several muscular diseases (*Cho et al., 2017*; *Janin and Gache, 2018*; *Norton and Phillips-Cremins, 2017*; *Puckelwartz et al., 2009*).

Proteins localized inside myonuclei such as Lamin A/C, Emerin, or Nuclear Envelope Proteins (NEPs), connect the nuclear lamina to the cytoskeleton through the 'LInker of Nucleoskeleton and Cytoskeleton' (LINC) complex, which plays a critical role in the maintenance of myonuclei shape and localization (*Chang et al., 2015*; *Lee and Burke, 2018*; *Levy et al., 2018*; *Mattioli et al., 2011*). Additionally, molecular motors and Microtubule-Associated Proteins (MAPs), in interaction with actin/microtubule networks, are also involved in myonuclei localization during muscle formation (*Casey et al., 2003*; *Gache et al., 2017*; *Metzger et al., 2012*; *Rivero et al., 2009*). Importantly, the specific localization and shape of myonuclei, determined by mechanical forces emanating from different cytoskeleton networks, act directly on chromatin organization and gene expression (*Chojnowski et al., 2015*; *Kirby and Lammerding, 2018*; *Ramdas and Shivashankar, 2015*; *Robson et al., 2016*). Yet, how myonuclei positioning in mature myofibers is set and how it regulates the signaling pathways maintaining myofiber integrity is still poorly understood.

To identify proteins involved in the maintenance of myonuclei positioning, we used an in vitro primary myotubes/myofibers culture system and identified MACF1 (Microtubule Actin Crosslinking Factor 1) as a regulator of myonuclei positioning during the late phases of myofiber maturation. MACF1 is a member of the spectraplakin family of proteins. It is a huge protein composed of multiple domains that allow its interaction with actin filaments and microtubules to stabilize them as adhesion structures in a variety of tissues (*Kodama et al., 2003*; *Sanchez-Soriano et al., 2009*). *MACF1* gene duplication (resulting in a decrease of MACF1 protein) is linked with a neuromuscular disease condition (*Jørgensen et al., 2014*) and *MACF1* variants are associated with congenital myasthenia (*Oury et al., 2019*). The *C. elegans* orthologous, VAB-10 and the *Drosophila* orthologous, Shot, are essential for nuclei migration in distal tip cells of the somatic gonad and myotendinous junction formation, respectively (*Bottenberg et al., 2009*; *Kim et al., 2011*; *Lee and Kolodziej, 2002*). Interestingly, Shot was proposed to contribute to the formation and maintenance of a perinuclear shield in muscle from drosophila larvae, contributing to myonuclei localization (*Wang and Volk, 2015*). However, the analysis of MACF1 functions in the control of extra-synaptic myonuclei position and muscle homeostasis is still pending.

Here, using in vitro and in vivo models, we report that MACF1 is important for the maintenance of myonuclei patterning in mammalian myofibers. MACF1 contributes to extra-synaptic myonuclei motion and positioning through the regulation of microtubules dynamics. We also reveal that MACF1 is essential for proper NMJ formation and mitochondria homeostasis.

## Results

### MACF1 uses microtubules to set myonuclei spreading during myofibers maturation

To identify new factors that contribute to myonuclei spreading in myofibers, we purified proteins able to bind to Taxol-stabilized microtubules from 3 days primary mouse myotubes and 13 days differentiated myofibers (*Figure 1A*). The major Microtubule-Associated-Proteins (MAPs) identified by mass-spectrometry using this protocol revealed the significant presence of MACF1 protein (*Figure 1—figure supplement 1A–B*). MACF1 is a member of the spectraplakin family known to play an architectural role through regulation of the spatial arrangement and function of specific organelles such as the nucleus, the mitochondria, the Golgi apparatus, and the sarcoplasmic reticulum (*Boyer et al., 2010*). MACF1 is a large protein mainly described as a cytoskeleton linker that binds and aligns microtubules and actin networks (*Preciado López et al., 2014*). In developing myofibers, actin is actively involved in myoblasts fusion and contributes to sarcomere formation (*Kim et al., 2015*; *Sanger et al., 2010*) while microtubules actively participate in myofiber elongation and myonuclei spreading (*Metzger et al., 2012*). Although MACF1 was initially described to be predominantly expressed in neurons and muscle (*Bernier et al., 2000*), its role in muscle fibers remains

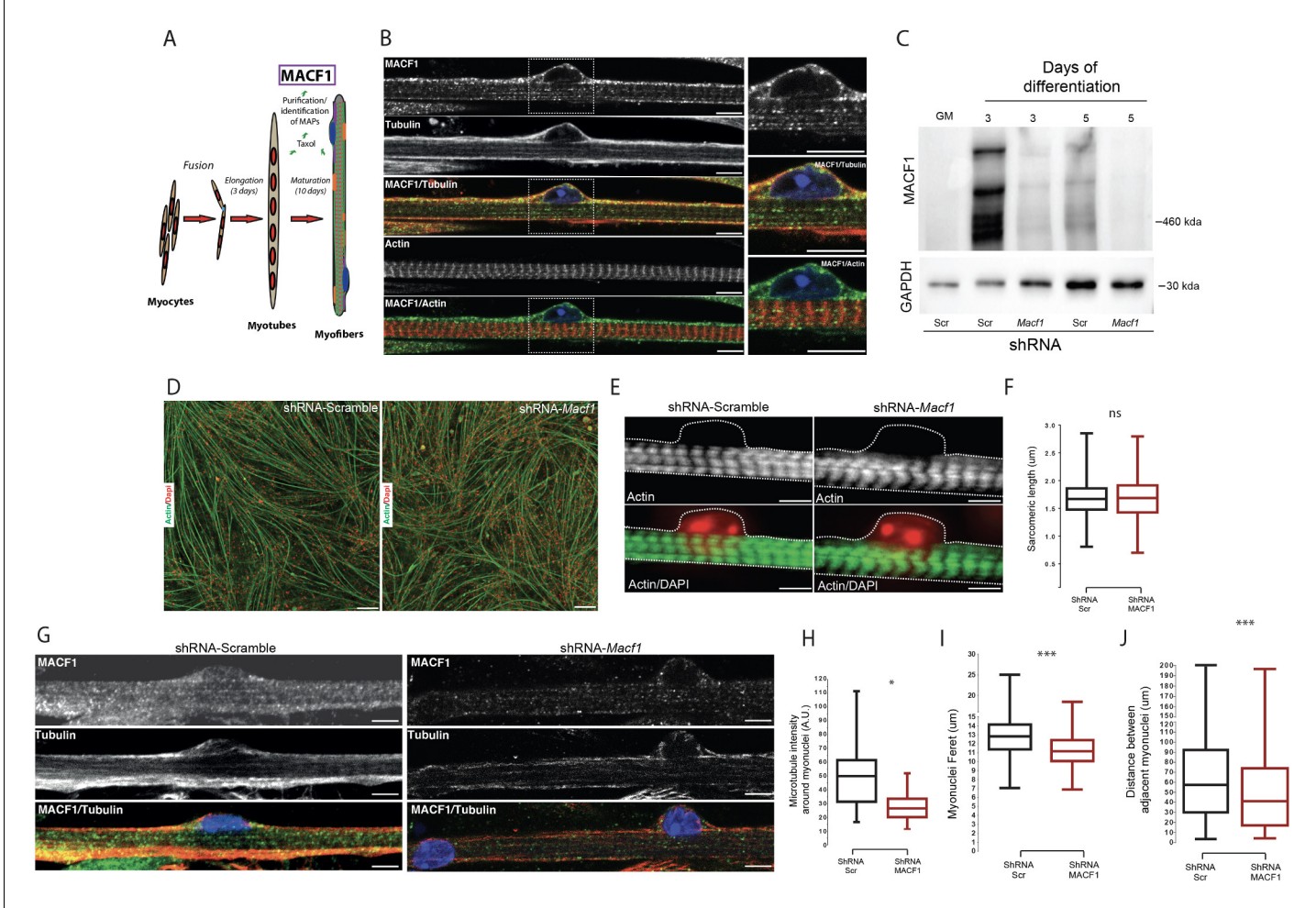

**Figure 1.** MACF1 is an essential regulator of myonuclei positioning in mature myofibers. (**A**) Scheme presentation of the sequential steps to obtain immature myotubes and mature myofibers from primary mouse myoblasts. Taxol stabilized microtubules proteome was collected either from WT elongated myotubes or from mature contractile myofibers. For downregulating experiments, siRNA/shRNA were transfected in early steps of differentiation just before myotubes formation. (**B**) Confocal immunofluorescence images (63×) presenting the organization of MACF1 (green) and total alpha tubulin or F-actin (red) in WT mature mouse primary myofibers. Scale bar = 10 μm. (**C**) Western blot analysis of MACF1 protein expression (main two isoforms at 830 and 630 kD) in total protein extracts of primary cells at proliferation cycle (GM for growth media) and in 3- or 5 days post-differentiation myotubes or myofibers. Cells were treated either with a scramble shRNA or a pool of four shRNAs targeting *Macf1*. GAPDH was used as loading control. (**D**) Immunofluorescence staining (10×) of F-actin (green) and myonuclei (red) in primary myofibers treated either with scramble shRNA (left panel) or with a pool of four distinct shRNAs targeting *Macf1* (right panel) after 13 days of differentiation. Scale Bar = 50 μm. (**E**) Representative immunofluorescence microscopy images (63×) of F-actin (green) and myonuclei (red) in primary myofibers treated with scramble shRNA (left panel) or with a pool of four distinct shRNAs targeting *Macf1* (right panel) after 13 days of differentiation. Scale Bar = 10 μm. (**F**) Sarcomere length measured on the immunofluorescence images of F-actin from 13 days mature primary myofibers treated with either scramble shRNA or a pool of four distinct shRNAs targeting *Macf1*. Data were obtained from three individual experiments. Line is set at median. Ns stands for not significant (Student's t test). (**G**) Confocal immunofluorescence images (63×) presenting the organization of total alpha tubulin (red) or MACF1 (green) in mature primary myofibers following treatment with either scramble shRNA (left panel) or a pool of four distinct shRNAs (right panel) targeting *Macf1*. Scale bar = 10 μm. (**H–J**) Mean values for fluorescent intensity of total alpha tubulin (**H**), myonuclei ferret (**I**) and distance between adjacent myonuclei (**J**) measured on mature primary myofibers following treatment with either scramble shRNA or a pool of four distinct shRNAs targeting *Macf1*. All panels were generated from data obtained from three individual experiments. Line is set at median. *p<0.05 and ***p<0.001 (Student's t test).

The online version of this article includes the following source data and figure supplement(s) for figure 1:

**Source data 1.** Table showing data from experiments plotted in *Figure 1F,H,I and J*.

**Source data 2.** Whole immunoblotting membrane for MACF1 and GAPDH plotted in *Figure 1C*.

**Figure supplement 1.** Reduction of MACF1 does not affects myotubes formation or myonuclei spreading within myotubes.

**Figure supplement 1—source data 1.** Table showing data from experiments plotted in *Figure 1—figure supplement 1D,F and G*.

**Figure supplement 1—source data 2.** Whole immunoblotting membrane for MACF1 and GAPDH plotted in *Figure 1—figure supplement 1C*.

**Figure supplement 2.** MACF1 is essential for myonuclei positioning and shape in mature myofibers.

*Figure 1 continued on next page*

*Figure 1 continued*

**Figure supplement 2—source data 1.** Table showing data from experiments plotted in *Figure 1—figure supplement 2A–C*.

elusive. We first addressed the localization of MACF1 in mature myofibers formed in vitro. Our immunofluorescence approach revealed a dotted pattern all along the myofibers that mainly colocalized with its cortical microtubule network, in agreement with our mass-spectrometry results (*Figure 1B*). Of importance, MACF1 accumulates preferentially around the peripheral myonuclei in mature myofibers (*Figure 1B*, zoom panels). Conversely, MACF1 is clearly excluded from actin network structures (*Figure 1B*).

To determine whether MACF1 is important for skeletal muscle development, we addressed its role during myotubes formation and in mature myofibers. Mouse *Macf1* mRNA was previously shown to increase steadily during myogenesis (*Leung et al., 1999*). We first observed an absence of MACF1 in proliferating myoblasts and confirmed a burst of MACF1 protein expression during the early steps of myotubes formation using primary mouse myoblasts cells and the C2C12 myogenic cell line (*Figure 1C*, *Figure 1—figure supplement 1C–D*). MACF1 expression was dramatically reduced upon transfection of siRNA or small hairpin RNA (shRNA) targeting the MACF1 mRNA compared to their respective scrambled (Scr) controls (*Figure 1C*, *Figure 1—figure supplement 1C*). MACF1 down-regulation had no impact on myotubes formation and architecture, as reflected by the unaffected ratio of myotube length to myonuclei content (*Figure 1—figure supplement 1E–F*). In accordance, neither myonuclei distribution as assessed by the mean distance between myonuclei and myotube's centroid nor the statistical distribution of myonuclei along myotubes length were affected following MACF1 down-regulation (*Figure 1—figure supplement 1G–I*). These results were confirmed in C2C12 myotubes cells where no alteration in myonuclei spreading was observed following MACF1 downregulation (*Figure 1—figure supplement 1J*).

To further investigate the implication of MACF1 in mature myofibers, primary mouse myotubes were maintained in differentiation media for 10 additional days using a protocol that allows myofibrillogenesis, sarcomeric structures formation and peripheral myonuclei adopting a flatten architecture and regular spreading along myofibers (*Falcone et al., 2014*). This long-term differentiation approach showed that MACF1 depletion did not impact the global maturation of myofibers, as reflected by homogeneous myofibers width and unaffected actin network striation (*Figure 1D–F*; *Figure 1—figure supplement 2A*). Yet, when addressing the global integrity of the microtubule network in these mature myofibers by immunofluorescence, we found a 50% decrease of the microtubule staining intensity in the vicinity of the myonuclei perimeter and an apparently less dense cortical network (*Figure 1G–H*). Accordingly, myonuclei shape was much rounder in MACF1-depleted condition, as appreciated by a significant reduction of the myonuclei feret diameter (*Figure 1I*, *Figure 1—figure supplement 2B*). As microtubules are used in developing myofibers by multiple molecular motors to determine myonuclei spreading (*Gache et al., 2017*), we investigated the impact of MACF1 depletion on this process and observed a significant reduction in the mean distance between adjacent myonuclei (*Figure 1J*, *Figure 1—figure supplement 2C*) in MACF1-depleted myotubes.

## MACF1 controls microtubules dynamics and myonuclei motion in developing muscle fibers

To better understand the localization of MACF1, as driven by its Microtubule Binding Domain (MTBD), we expressed only the MTBD of MACF1 in immature primary myotubes and in mature primary myofibers (*Figure 2A–B*). Interestingly, in primary myotubes, MACF1-MTBD appeared as small comets that merged with the microtubule network all along myotubes length (*Figure 2A*). In contrast, in mature myofibers, we observed a strong accumulation of stable MACF1-MTBD longitudinal bundles close to myonuclei, in addition to small comets dispersed along the myofibers (*Figure 2B*).

MACF1 was previously shown to regulate microtubules polymerization (*Alves-Silva et al., 2012*; *Ka et al., 2016*; *Wang et al., 2015*). We thus questioned, in our system, how microtubule dynamics evolve during the myofibers maturation steps and whether MACF1 depletion impacts this process. To study the microtubule network dynamics in developing myofibers, we used the tracking of a fluorescently-tagged End Binding protein member (EB3) at different time points during maturation of in

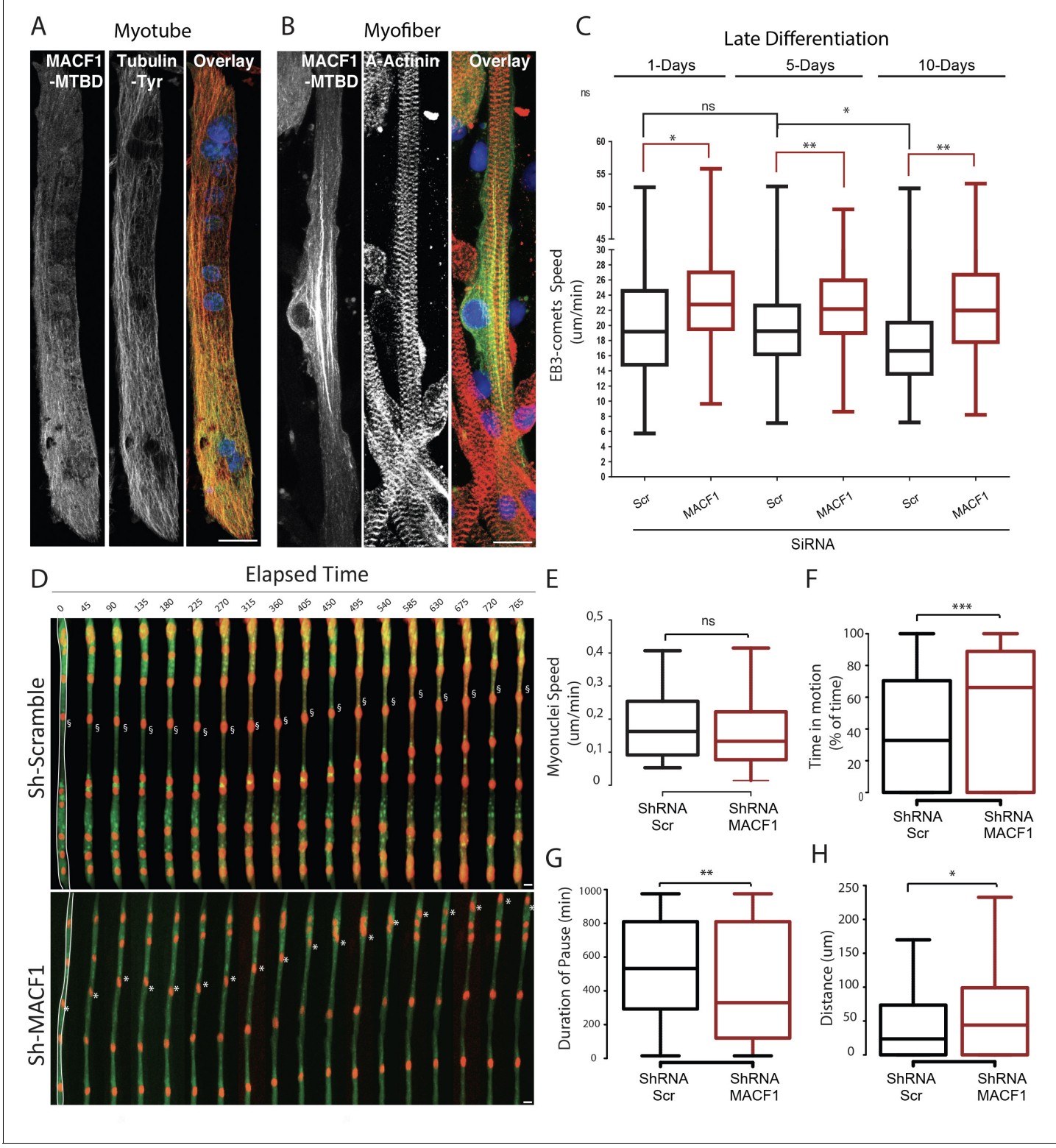

**Figure 2.** MACF1 regulates myonuclei dynamics through the control of microtubules. (**A–B**) representative confocal immunofluorescence images (63×) presenting MACF1-MTBD (green) and tyrosinated alpha tubulin (red) in primary myotubes (**A**) or alpha actinin (red) in primary myofibers (**B**). Scale bar = 10 μm. (**C**) Quantification of EB3 comets speed in primary cells treated with either scramble siRNA or a pool of 3 individual siRNAs targeting *Macf1* in primary myotubes at three different time points during maturation. At least 200 comets from three individual experiments were mapped per condition. Line is set at median. Ns stands for not significant, *p<0.05 and **p<0.01 (Student's t test). (**D**) Frames from a 14 hr time-lapse movie of two channels

*Figure 2 continued on next page*

*Figure 2 continued*

(shRNA in green and lamin-chromobody in red) taken 5 days post differentiation from primary myotubes treated with either a scramble shRNA (upper panel) or a mix of shRNAs targeting *Macf1* (lower level). In the first frame (on the left), myofibers are selected in white, which corresponds to the region used to create the adjacent kymograph. Scale bar = 10 μm. (E–H) Speed (E), Time in motion (F), Duration of pauses (G), and Distance travelled by myonuclei (H) were quantified using SkyPad analysis (*Cadot et al., 2014*). Quantifications were done on myonuclei within different myofibers from three individual experiments. Line is set at median. Ns stands for not significant, *p<0.05, **p<0.01, and ***p<0.001 (Student's t test).

The online version of this article includes the following source data for figure 2:

**Source data 1.** Table showing data from experiments plotted in *Figure 2C and E–H*.

vitro primary myofibers (*Figure 2C*, *Video 1*). During myofibers maturation process, EB3-comets speed was relatively constant (≈ 19 μm/min) until 5 days of maturation. After this time point, EB3-comets speed decreased significantly, reflecting a progressive downturn in microtubule dynamics (*Oddoux et al., 2013*; *Figure 2C*). However, the velocity of EB3 comets was increased by 15% upon MACF1 downregulation and remained high (≈ 23 μm/min) all along the process of myofibers maturation (*Figure 2C*). This is consistent with a role for MACF1 in the regulation of microtubule polymerization, mainly as a stabilization factor. As myonuclei preferentially use the microtubule network to move longitudinally along myotubes (*Gache et al., 2017*; *Metzger et al., 2012*), we next investigated the impact of the increase in microtubule dynamics on myonuclei movements during myofibers maturation. Just before fusion, myoblasts were co-transfected with the lamin-chromobody to visualize myonuclei concomitantly with a shRNA targeting either a scramble sequence or *Macf1*, and a GFP reporter (*Figure 2D*, *Video 2*). We tracked myonuclei in 5-day-differentiated myofibers every 15 min for 14 hr and analyzed their displacement parameters by the SkyPad method (*Cadot et al., 2014*). In control conditions, myonuclei spent 32% of the time in motion at a median speed of 0.16 μm/min, resulting in a displacement of 24 μm after 14 hr (more than twice bigger than the mean myonuclei feret). Interestingly, in the absence of MACF1, the median velocity of myonuclei was not significantly changed (*Figure 2E*) but the percentage of time that myonuclei spent in motion was doubled, reaching 66% of the time (*Figure 2F*). Accordingly, the median motion pause was 530 min in the control conditions while it fell down to 330 min in the absence of MACF1 (*Figure 2G*). Overall, in the absence of MACF1, myonuclei traveled twice more distance than in control conditions with a mean distance of 44 μm in 14 hr (*Figure 2H*).

Thus, depletion of MACF1 generates increased microtubule dynamics, which in turn elevate myonuclei motion during myofibers maturation, inducing failure in myonuclei spreading in mature myofibers.

## Conditional MACF1-KO adult mice show microtubule network disorganization in their skeletal muscle fibers

To further investigate the role of MACF1 in skeletal muscle development, we used a conditional *Macf1* knockout mouse line in which exons 6 and 7 of the *Macf1* gene are floxed (*Goryunov et al., 2010*). Homozygous mice were crossed with mice carrying the Cre-recombinase expression under the control of the Human Skeletal muscle Actin (HSA) promoter (*Miniou, 1999*). Briefly, mice carrying two floxed alleles of *Macf1* (*Macf1*[f/f]) were crossed with heterozygous mice for *Macf1* (*Macf1*[f/−]) that carried the HSA::Cre transgene (HSA::Cre; *Macf1*[f/−]), generating the conditional

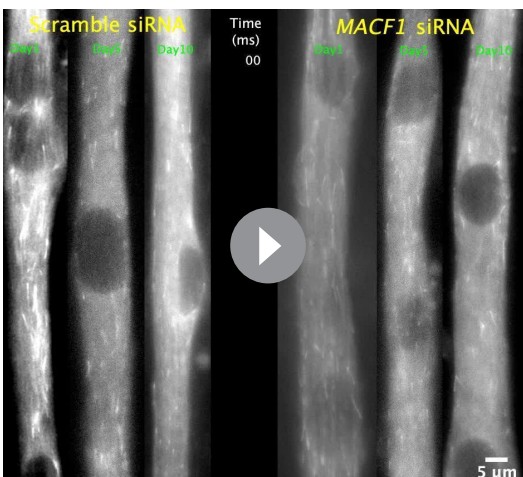

**Video 1.** Time-lapse experiments of primary myoblasts co-transfected with EB3-GFP plasmid and Scramble-siRNA at 1 day, 5 days, and 10 days post 'late differentiation' initiation (left panels) or primary myoblasts co-transfected with EB3-GFP plasmid and *Macf1*- siRNA at 1 day, 5 days, and 10 days post 'late differentiation' initiation (right panels). Each frames represent 500 milliseconds and was recorded for a period of time of 7 s.

https://elifesciences.org/articles/70490#video1

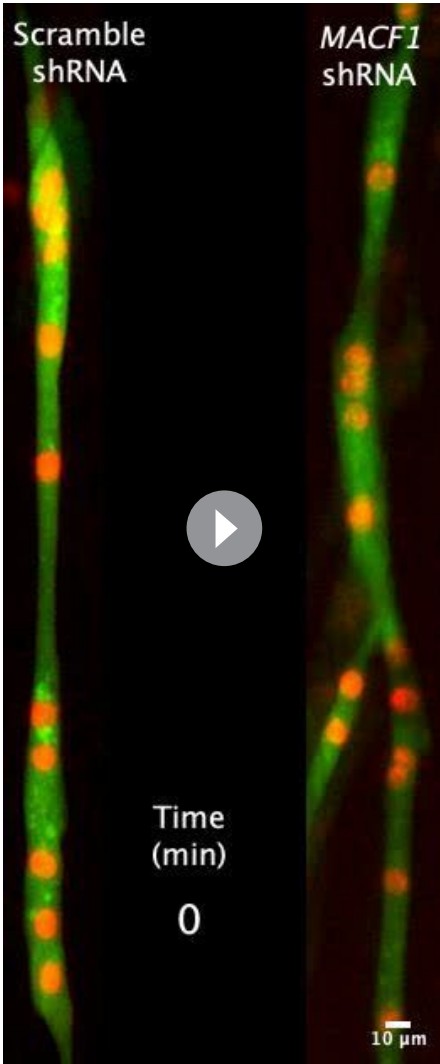

**Video 2.** Time-lapse experiments of primary myoblasts co-transfected with Lamin-Chromobody-RFP plasmids and either Scramble shRNA (left panel) or a pool of four individual *Macf1* shRNA (rigth panel). Time laps images were captured form myotubes 5 days after starting differentiation process. Primary myotubes were recorded every 15 min for a period of time of 14 hr. https://elifesciences.org/articles/70490#video2

mutant (HSA::Cre; *Macf1*^f/f^; hereafter referred to as *Macf1* Cre+) and control mice (WT; *Macf1*^f/f^ referred as *Macf1* Cre-) (*Figure 3A–B*). We quantified the efficiency of this approach by measuring the loss of MACF1 mRNA in muscle lysates and observed that its levels were reduced by 70% (*Figure 3C*, *Figure 3—figure supplement 1A*). Besides, MACF1 downregulation was confirmed by western blotting (*Figure 3—figure supplement 1B*). *Macf1* conditional-KO mice were born at Mendelian frequency and appeared indistinguishable from WT littermates (data not shown). Comparing the body weight evolution between conditional-KO (*Macf1* Cre+) and control mice (*Macf1* Cre-) throughout growing and maturation did not reveal any difference between the two groups (*Figure 3—figure supplement 1C–E*).

We first addressed the impact of MACF1 depletion on the cytoskeleton network using mature myofibers extracted from the *Tibialis Anterior* muscles of WT and conditional MACF1-KO mice. As expected from our in vitro experiments, immunofluorescence approach confirmed the preferred co-localization of MACF1 with the microtubule network rather than with the actin network (*Figure 3D*). We also observed a clear accumulation of MACF1 in the vicinity of myonuclei (*Figure 3D*, arrows) as previously reported in *Drosophila* myofibers but not in murine myofibers (*Oury et al., 2019*; *Wang et al., 2015*). In *Macf1* Cre+ muscle myofibers, MACF1 staining was clearly less intense all along the myofibers and particularly within the myonuclei vicinity. Also, the microtubule network organization was clearly altered in *Macf1* Cre+ myofibers while there was no concurrent sign of actin network defect (*Figure 3D*). The preserved integrity of the actin network was confirmed by the absence of change in sarcomere length at different ages (*Figure 3E*, *Figure 3—figure supplement 1F*). To quantify the spatial disorganization of the microtubule network, we used a texture detection tool (TeDT) (*Liu and Ralston, 2014*). Our data show that the longitudinal microtubule network is the most affected regardless of the age of the animals (2-, 6-, or 12-month-old mice (*Figure 3F*, *Figure 3—figure supplement 1G–H*)), in agreement with the loss of MACF1 staining in the *Macf1* Cre+ muscle myofibers (*Figure 3D*). Altogether, these evidences demonstrate that MACF1 stabilizes the microtubule network during myofibers maturation.

MACF1 can modulate microtubule dynamics through the interaction with different partners such as CLAPS2, EB1, CAMSAP3, Nesprin, Map1B, ErbB2 (*Ka et al., 2014*; *Noordstra et al., 2016*; *Pereira et al., 2006*; *Ryan et al., 2012*; *Zaoui et al., 2010*). In addition, microtubules are subject to a variety of post-translational modifications (PTMs) that are associated with changes in microtubules dynamics (*Nieuwenhuis and Brummelkamp, 2019*). Among these, tubulin de-tyrosination is associated with longer-living microtubules, whereas dynamic microtubules are mainly tyrosinated (*Bulinski and Gundersen, 1991*). To further investigate the role of MACF1 in the long-term balance of microtubule dynamic in myofibers, we addressed the microtubule tyrosination/de-tyrosination

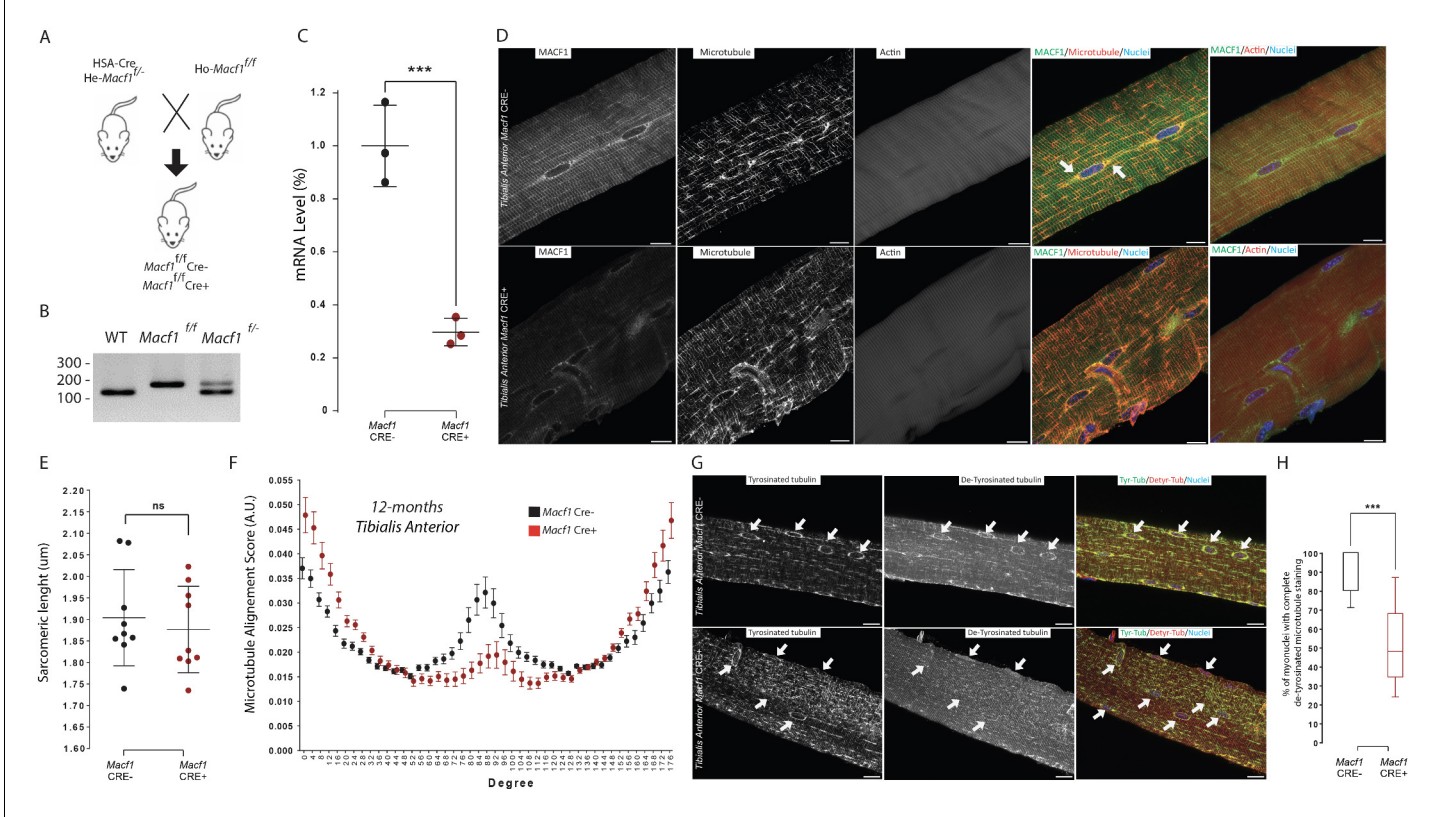

**Figure 3.** MACF1 is necessary for global and perinuclear microtubules organization within skeletal muscles in vivo. (**A**) Scheme of the strategy used to generate muscle specific *Macf1* knockout mouse model. (**B**) Representative PCR gel confirming the floxed *Macf1* gene in mutant mice carrying Cre transgene. (**C**) Quantification of efficacy of the Cre-lox system in our muscle specific mouse model by analyzing the *Macf1* gene expression level. qRT-PCR was carried out on total mRNA extracted from *Gastrocnemius* of 3 WT and 3 *Macf1* KO mice of 12-month-old. Following the calculation of relative gene expression level on housekeeping genes for each mouse, the mean expression level of *Macf1* was calculated for the WT mice. To generate the graph, the *Macf1* expression level of each WT and KO mouse was then normalized on the previously calculated mean value. Line is set at mean (with SEM). ***p<0.001 (Student's t test). (**D**) Representative confocal images (63×) of isolated myofibers from *Tibialis Anterior* of *Macf1* Cre- (upper panel) and Cre+ (lower panel) mice, showing MACF1 (green), myonuclei (blue) and microtubules or F-actin (red). Scale Bar = 15 µm. (**E**) Sarcomere length quantified from the alpha actinin staining from 3 *Macf1* Cre- and 3 *Macf1* Cre+ 12-month-old mice. Following myofibers isolation from *Tibialis Anterior*, the distance between alpha actinin transverse bands of 3 myofibers per mouse was measured. Line is set at mean (with SD). Ns stands for not significant (Mann-Whitney test). (**F**) Microtubules network organization analysis of at least three myofibers from each of the three WT and three conditional *Macf1*-KO mice using TeDT software. The final generated graph presents a global score for each given degree of microtubules orientation, with 0 and 180 degrees corresponding to the longitudinal microtubules and 90 degrees corresponding to the transverse microtubules. Mean values (with SEM) is presented. (**G**) Representative confocal images (63×) from myofibers isolated from *Tibialis Anterior* issued from 12-month-old *Macf* Cre- (upper panel) or Cre+ (lower panel) mice, presenting tyrosinated tubulin (green), de-tyrosinated tubulin (red) and myonuclei (blue). White arrows point to myonuclei. Scale Bar = 15 µm. (**H**) Quantification of the percentage of myonuclei presenting the total de-tyrosinated tubulin ring. Quantifications were done on confocal images obtained from several isolated myofibers from *Tibialis Anterior* of three WT and three conditional *Macf1*-KO mice at the age of 12 months. Line is set at median. ***p<0.001 (Student's t test).

The online version of this article includes the following source data and figure supplement(s) for figure 3:

**Source data 1.** Table showing data from experiments plotted in *Figure 3C,E,F and H*.

**Figure supplement 1.** MACF1 conditional KO mice present progressive microtubules disorganization.

**Figure supplement 1—source data 1.** Table showing data from experiments plotted in *Figure 3—figure supplement 1A and C–I*.

status in muscles from our conditional-KO and control mice (*Figure 3G–H*). In *Macf1* Cre- muscle fibers, tyrosinated tubulin is found all along microtubules and in the vicinity of myonuclei along myofibers. Conversely, de-tyrosinated tubulin was found preferentially in the vicinity of myonuclei. While in *Macf1* Cre+ muscles, there was no change in the tyrosinated tubulin pattern, de-tyrosinated tubulin staining at myonuclei periphery was either severely reduced or absent (*Figure 3G*, arrows). Interestingly, this modification was relevant only in muscles from 12-month-old *Macf1* Cre+ mice

(*Figure 3H*, *Figure 3—figure supplement 1I*), suggesting a progressive impact of MACF1 loss on the microtubules pattern. Therefore, MACF1 plays a critical role in the long-term maintenance of stability of the peri-nuclear and longitudinal microtubule network of myofibers.

## Muscle-specific conditional MACF1-KO mice exhibit progressive myonuclei mislocalization

We next assessed myonuclei localization along muscle fibers of *Macf1* Cre+ mice. *Tibialis Anterior* myofibers were extracted from 2-, 6-, and 12 months old *Macf1* Cre- and *Macf1* Cre+ mice and immunofluorescence staining was used to analyze myonuclei distribution (*Figure 4A*). We found that the shortest distance between neighboring myonuclei was increasing throughout development in wild-type muscles, reaching a plateau at 6 months of age (*Figure 4B*). In MACF1 depleted myofibers, not only this increase was not observed during development but there was also a significant drop in the value for the distance between myonuclei in myofibers from 12-month-old mice (*Figure 4B*). In addition, the increase in the myonuclei feret value observed in control myofibers, was abolished in *Macf1* Cre+ myofibers (*Figure 4C*), leading to an increase in myonuclei roundness (*Figure 4D*), consistent with the slackening of the cytoskeleton pressure on the myonuclei membrane (*Wang et al., 2015*).

In normal conditions, myonuclei are regularly spaced at the periphery along muscle fibers (*Bruusgaard et al., 2006*). We compared myonuclei localization in different types of muscles from *Macf1* Cre- and *Macf1* Cre+ mice by analyzing muscle cross-sections stained with DAPI and laminin (to determine the limit of each myofiber) (*Figure 4E*, *Figure 4—figure supplement 1A*). We observed a significant increase in the number of delocalized myonuclei in the *Macf1* Cre+ muscles, either at the center or elsewhere dispatched within the myofibers of several skeletal muscles (*Tibialis Anterior*, *Soleus* and *Gastrocnemius*) in 12-month-old mice (*Figure 4E–F*, *Figure 4—figure supplement 1A–B*). Interestingly, no significant delocalization of myonuclei was observed in 3- and 8-month-old mice (*Figure 4—figure supplement 1A–B*), suggesting that peripheral myonuclei alteration precede myonuclei internalization. Remarkably, peripheral myonuclei spreading is altered concomitantly with microtubules disorganization, while myonuclei internalization correlates with the disappearance of stable microtubules around myonuclei. With respect to myofiber size, no significant alteration was detected in the cross-sectional area (CSA) in *Tibialis Anterior* muscles from 3-month-old mice (*Figure 4G*). Nonetheless, we observed a shift of the distribution towards a smaller CSA in muscles from 8-month-old *Macf1* Cre+ mice, which was maintained at 12 months of age (*Figure 4G*). Of note, changes in myonuclei positioning were not associated with an alteration in the total number of myofibers per muscle (*Figure 4—figure supplement 1C*), nor with a massive regeneration process, as evidenced by the absence of *Myogenin* or *MyoD* mRNA upregulation (*Figure 4—figure supplement 1D–E*). Thus, MACF1 is implicated in the maintenance of peripheral myonuclei spreading in adult myofibers as well as in the prevention of myonuclei internalization.

## MACF1-deficient muscles exhibit progressive neuromuscular junction alteration

MACF1 contributes to the maintenance of the neuromuscular junction (NMJ) by recruiting microtubules at that location (*Oury et al., 2019*) where synaptic myonuclei are clustered underneath the junction (*Bruusgaard et al., 2006*). We thus investigated the potential implication of MACF1 in myonuclei clustering at the NMJ. Immunofluorescence staining revealed no clear accumulation or specific localization of MACF1 at the NMJ, in contrast with a previous report on mouse muscle (*Oury et al., 2019*). However, consistent with observations from this group, there was a clear progressive alteration of acetylcholine receptors (AChRs) patterning in myofibers from *Macf1* Cre+ mice, characterized by a decrease of AChRs clusters size and an increase of NMJ fragmentation (*Figure 5A–C*). This AChRs clustering impairment was confirmed in different muscles (*Tibialis Anterior*, *Extensor Digitorum Longus, and Rectus Lateralis*) (*Figure 5—figure supplement 1A–B*). We also confirmed these results on the neuromuscular junction of *Drosophila* model, using the muscle-specific driver *(Mef2-GAL4)* to express RNAi against *Drosophila Shot* (*Macf1* orthologous) (*Wang et al., 2015*). Immunostaining against Shot confirmed a preferential localization of the protein around myonuclei but not at the NMJ and showed a significant reduction of protein level around myonuclei in Shot-KD larval muscles (*Figure 5—figure supplement 1C–D*). In accordance with the conditional-KO mouse model,

we found that the size of active zones of synaptic buttons and post-synaptic terminals were decreased in Shot-KD larval muscles (*Figure 5—figure supplement 1E–F*).

We next studied the association of synaptic myonuclei with the NMJ. We observed a progressive recruitment of myonuclei underneath the NMJ in *Tibialis Anterior* myofibers of control mice with age, concomitant with an enlargement of the area occupied by AChRs. In control mice, the number of synaptic myonuclei increased from approximately 7 at 2 months of age to 11 at 12 months of age (*Figure 5D*). In MACF1-depleted myofibers (*Macf1* Cre+), there was no new recruitment of synaptic myonuclei over the same period and the number of synaptic myonuclei remains close to 7 from 2- to 12 months of age (*Figure 5D*). As we identified that microtubule network is altered along muscle fibers in MACF1-depleted condition, we analyzed the potential changes in tyrosinated/de-

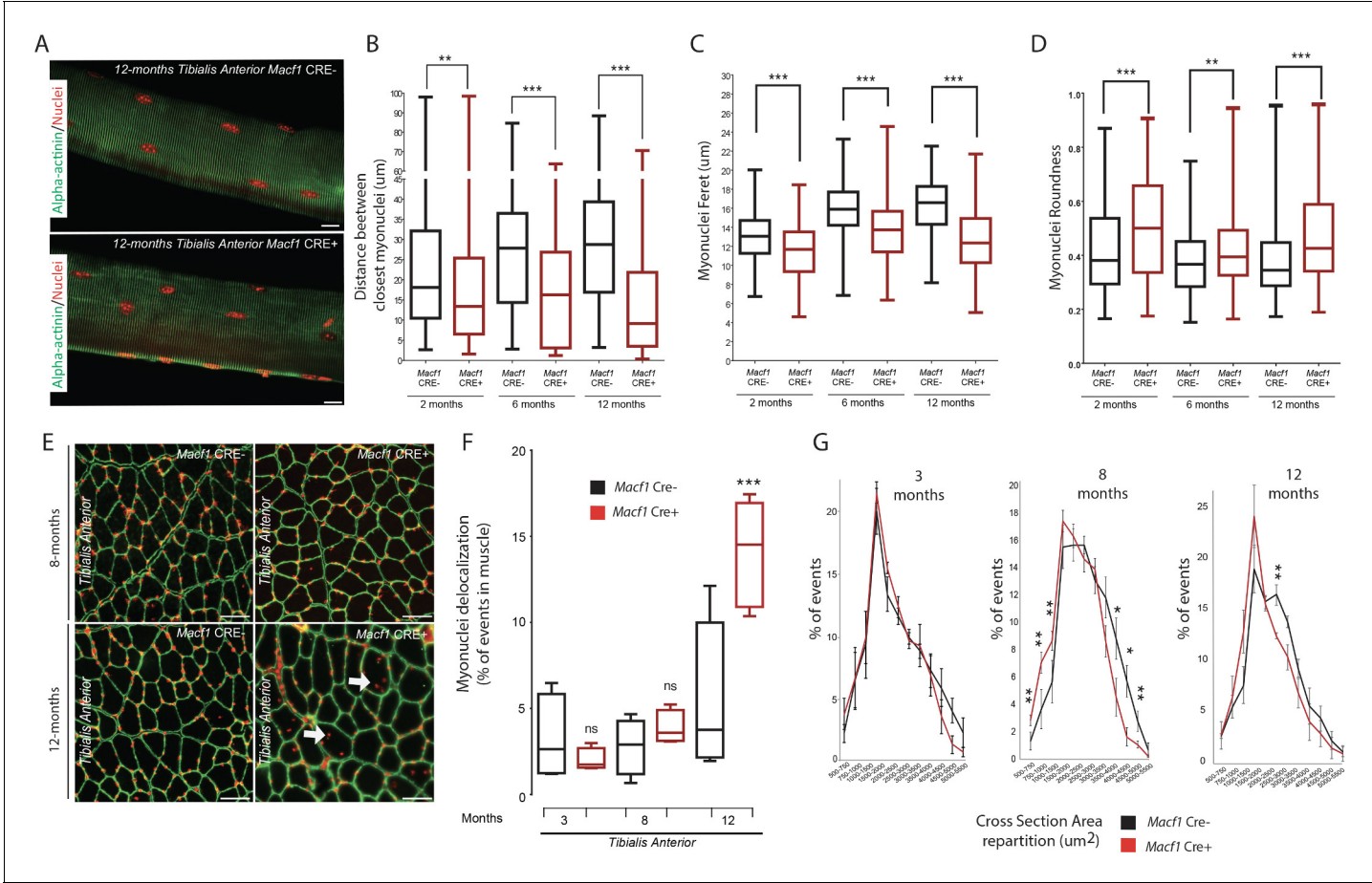

**Figure 4.** Loss of MACF1 alters myonuclei shape and positioning progressively with age. (A) Representative confocal images (63×) of myofibers isolated from *Tibialis Anterior* of *Macf1* Cre- (upper panel) and Cre+ (lower panel) mice at the age of 12 months presenting alpha-actinin (green) and myonuclei (red). Scale Bar = 15 µm. (B–D) Quantification of distance between the nearest myonuclei (B), myonuclei feret (C) and myonuclei roundness (D) measured on isolated myofibers from *Tibialis Anterior* of *Macf1* Cre- and Cre+ mice (as presented in panel A) at 2-, 6-, and 12 months of age. Several myofibers were subjected to analysis per mouse. Each group gathers data obtained from myofibers of 3 or more mice. Line is set at median. **p<0.01 and ***p<0.001 (Student's t test). (E) Representative images of *Tibialis Anterior* muscle cross-section from *Macf1* Cre- (left panels) and *Macf1* Cre+ (right panels) mice at the age of 8 and 12 months, stained for myonuclei (red) and Laminin (green). Examples of myofibers with mis-localized myonuclei are indicated by white arrows. Scale Bar = 150 µm. (F) Quantification of the percentage of myofibers with mis-localized myonuclei in *Tibialis Anterior* of 3, 8, and 12 months-old *Macf1* Cre- and *Macf1* Cre+ mice. Line is set at median. Ns stands for not significant and ***p<0.001 (Student's t test). (G) Distribution of myofibers cross-section area in the *Tibialis Anterior* from 3-, 8-, and 12-months-old *Macf1* Cre- and *Macf1* Cre+ mice. Line is set at mean (with SEM). Ns stands for not significant, *p<0.05 and **p<0.01 (Student's t test).

The online version of this article includes the following source data and figure supplement(s) for figure 4:

**Source data 1.** Table showing data from experiments plotted in *Figure 4B–D and F–G*.

**Figure supplement 1.** Mis-localization of myonuclei in MACF1 conditional KO mice is independent from muscular regeneration.

**Figure supplement 1—source data 1.** Table showing data from experiments plotted in *Figure 4—figure supplement 1B–E*.

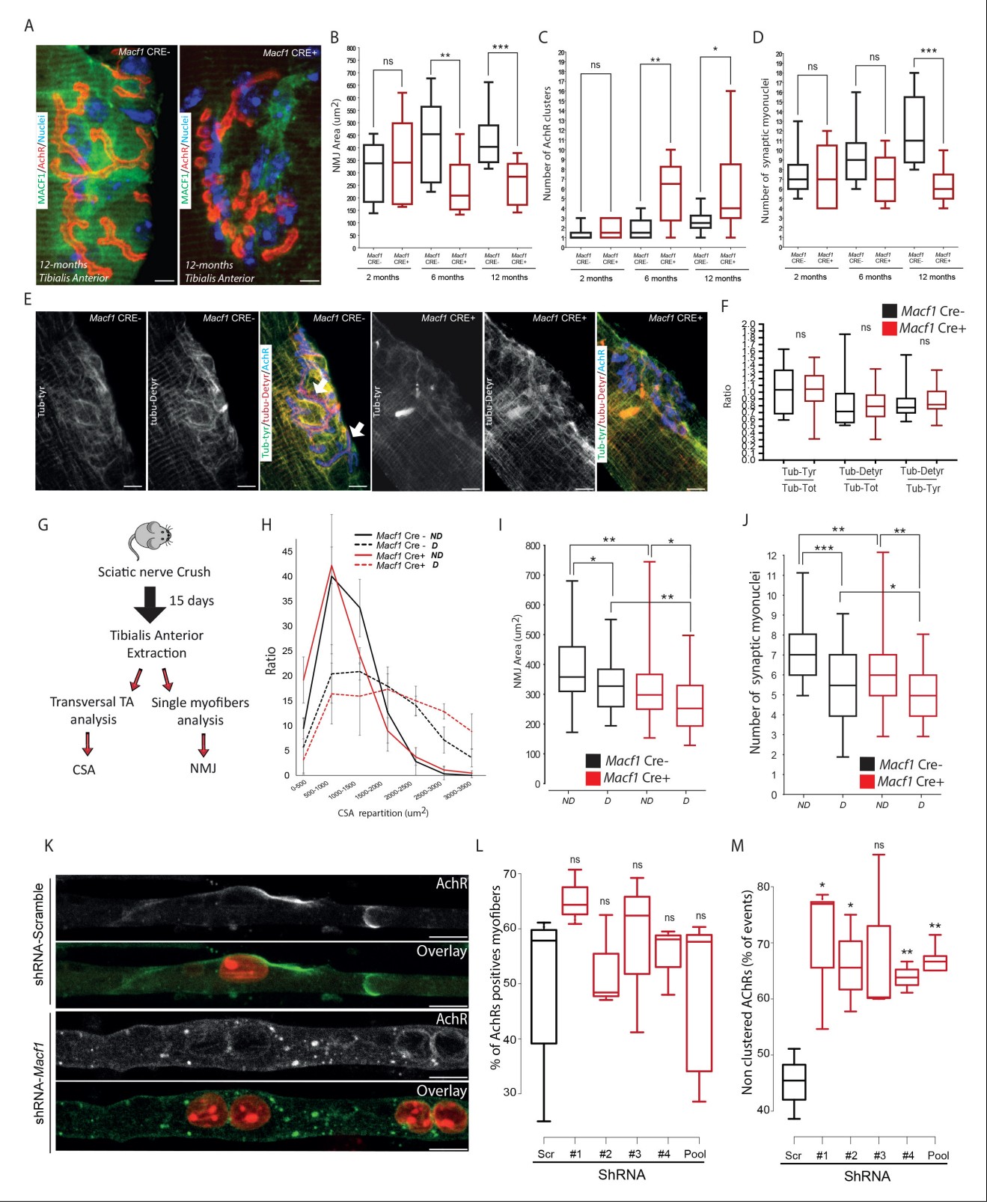

**Figure 5.** MACF1 controls AChRs clustering but has no impact on synaptic myonuclei at the NMJ. (**A**) Representative confocal images (63×) of post-synaptic NMJs within isolated myofibers from *Tibialis Anterior* of *Macf1* Cre- (left panel) and Cre+ (right panel) mice at the age of 12 months presenting MACF1 (green), AChRs clusters (red) and myonuclei (blue). Scale Bar = 10 μm. (**B–D**) Quantification of post-synaptic NMJs area (**B**), number of AChRs clusters (**C**) and number of synaptic myonuclei (**D**) measured on myofibers isolated from *Tibialis Anterior* of *Macf1* Cre- and Cre+ mice (as presented in

*Figure 5 continued on next page*

*Figure 5 continued*

panel A) at 2-, 6-, and 12 months of age. Several myofibers were subjected to analysis per mouse and each group is formed from at least three mice. Line is set at median. Ns stands for not significant, *p<0.05, **p<0.01 and ***p<0.001 (Student's t test). (E) Representative confocal images (63×) presenting tyrosinated tubulin (green), de-tyrosinated tubulin (red), and AChRs clusters (blue) in myofibers isolated from *Tibialis Anterior* of *Macf1* Cre- (three left panels) and Cre+ (three right panels) mice at the age of 12 months. Scale Bar = 10 µm. (F) Quantification of the ratios of fluorescence intensity from tyrosinated tubulin to total tubulin, de-tyrosinated tubulin to total tubulin and de-tyrosinated tubulin to tyrosinated tubulin. The mean fluorescence intensity of each given staining was measured specifically for the post-synaptic NMJs area (as presented in panel E) on NMJs of myofibers isolated from *Tibialis Anterior* of *Macf1* Cre- and Cre+ mice at the age of 12 months. Line is set at median. Ns stands for not significant (Student's t test). (G) Scheme of the strategy used to induce sciatic nerve crush and subsequent analysis of muscle fibers from *Macf1* Cre- and Cre+ mice. (H) Distribution of myofibers cross-section areas from non-denervated (referred as ND) and denervated (referred as D) *Tibialis Anterior* muscles of *Macf1* Cre- and Cre+ mice at the age of 8 months. Line is set at mean (with SEM). (I–J) Quantification of post-synaptic NMJs area (I) and number of synaptic myonuclei (J) measured on isolated myofibers from non-denervated (referred as ND) and denervated (referred as D) *Tibialis Anterior* muscles of *Macf1* Cre- and Cre+ mice at the age of 8 months. Line is set at median. *p<0.05, **p<0.01, and ***p<0.001 (Student's t test). (K) Representative fluorescence images (63×) of AChRs clusters (green) and myonuclei (red) in myofibers treated with scramble shRNA (two upper panels) or with a pool of the four distinct shRNAs targeting *Macf1* (two lower panels) after 10 days of differentiation in presence of Agrin in culture medium. Scale Bar = 10 µm. (L) Percentage of myofibers expressing immunofluorescence staining for AChRs in primary myofibers treated with scramble shRNA, with each of the four distinct shRNAs or with a pool of these four distinct shRNAs targeting *Macf1* after 10 days of differentiation in the presence of Agrin. Data are from three individual experiments. Line is set at median. Ns stands for not significant (Student's t test). (M) AChRs clustering distribution in cells treated with scramble shRNA, with each of the four individual shRNAs or with a pool of the four individual shRNAs targeting *Macf1* after 10 days of differentiation in the presence of Agrin. Data was collected from three individual experiments for each condition. Line is set at median. Ns stands for not significant, *p<0.05 and **p<0.01 (Student's t test).

The online version of this article includes the following source data and figure supplement(s) for figure 5:

**Source data 1.** Table showing data from experiments plotted in *Figures 5B–D,F,H–J and L–M*.

**Figure supplement 1.** MACF1 orthologue, Shot, is essential for NMJs organization in *Drosophila*.

tyrosinated tubulin distribution at the NMJ. Surprisingly, tubulin accumulation was not observed around the synaptic myonuclei of control myofibers, in contrast with the pattern observed at extra-synaptic nuclei (*Figure 5E*, *Figure 3G*). Instead, we detected tubulin accumulation close to the NMJ site of MACF1 mutant mice with a blurry patterning. Nonetheless, de-tyrosinated tubulin accumulation appeared highly correlated with AChRs staining even when AChR clusters were fragmented in the conditional-KO mice (*Figure 5E*, arrows). Finally, the ratio of both tyrosinated and de-tyrosinated tubulin vs total tubulin at NMJs were not different between *Macf1* Cre- and *Macf1* Cre+ mice (*Figure 5F*). This suggests that the role of MACF1 at the NMJ is not related to myonuclei clustering nor to the control of microtubule dynamics but is instead associated with AChRs clustering.

To further check that synaptic myonuclei location does not depend on MACF1, we used a sciatic nerve crush protocol to induce denervation (*Figure 5G*; *Milanic et al., 1999*). Using this technique, we clearly observed atrophy of the muscle fibers together with a reduction in the AChRs area in both *Macf1* Cre- and *Macf1* Cre+ conditions (*Figure 5H–I*). Interestingly, 15 days following denervation, the similar number of synaptic myonuclei per NMJ was lost in both *Macf1* Cre- and *Macf1* Cre+ conditions (*Figure 5J*), confirming that MACF1 is not directly implicated in the maintenance of synaptic myonuclei.

As the role of MACF1 at the NMJ seemed specifically related to AChR clustering, we tested whether MACF1 depletion in myofibers in vitro would fail to precociously cluster AChRs. For this, agrin-supplemented culture medium was used during in vitro differentiation of myofibers as this is known to induce clustering of AChRs (*Lin et al., 2005*; *Schmidt et al., 2012*; *Vilmont et al., 2016*; *Figure 5K*). Using immunofluorescence staining, we found that, even though the proportion of shRNA-MACF1-transfected myofibers expressing AChRs was not changed (*Figure 5L*), they displayed a 20% drop in the number of those presenting AChR clusters with a diameter larger than 8 µm (*Figure 5M*). This demonstrates that MACF1 is required for efficient AchR clustering in the myofiber.

Altogether, MACF1 is required to maintain the neuromuscular synapse integrity through AChRs clustering in developing and differentiated myofibers, which in turn controls synaptic myonuclei recruitment at the NMJ.

## Muscle fibers from MACF1-KO mice show muscle excitability defects and T-tubules alteration with preserved excitation-contraction coupling

To determine whether the above-described alterations impact muscle function, we tested muscular performance in young (4-month-old) and in adult (12-month-old) *Macf1* conditional-KO mice compared to age-matched control animals. The formers are mainly characterized by peripheral extra-synaptic myonuclei disorganization, while the latters also exhibit large internalization of extra-synaptic myonuclei mice. We used a strictly non-invasive experimental setup offering the possibility to stimulate the hindlimb muscles in vivo to record force production. Force was measured in response to incremental stimulation frequencies (from 1 to 100 Hz) to obtain the force-frequency relationship. There was no significant difference in the values for maximum force produced between the two groups, irrespective of the mouse age. However, in young mice, there was a large rightward shift of the force production capacity in conditional *Macf1* KO mice (*Figure 6A*), suggesting alteration of either the neuromuscular transmission or of the excitation-contraction coupling process. Of note, this shift was also maintained in adult mice, although in a less pronounced manner (*Figure 6A*).

In skeletal muscle, triads consist of one transverse tubule (T-tubule) with two appended terminal cisternae of sarcoplasmic reticulum (SR). The triad is the structure where excitation-contraction coupling (ECC) takes place. Efficient triad formation has been linked to microtubule organization in muscle cells (*Osseni et al., 2016*). Since *Macf1* conditional-KO myofibers exhibit microtubule disorganization, we used electron microscopy to study the ultrastructure of muscle in *Macf1* Cre+ mice and analyzed T-tubules and SR organization (*Figure 6B*). We found signs of alteration of the T-tubules in some *Macf1* Cre+ myofibers (*Figure 6B*, white arrows) consistent with a possible alteration of ECC. To test this possibility, we compared voltage-activated SR $Ca^{2+}$ release in *Flexor Digitalis Brevis* (FDB) muscle fibers isolated from conditional *Macf1* Cre+ mice and control mice. Confocal staining using di-8-anepps showed an overall comparable structure of T-tubule network in *Macf1* Cre- and in *Macf1* Cre+ FDB muscle fibers (*Figure 6C* -top and -middle panels). However, some isolated muscle fibers from the conditional mutant mice exhibited strong alteration of T-tubule orientation without perturbation in T-tubules density or sarcomere length (*Figure 6C* -Bottom panel, D-F). We next tested if SR $Ca^{2+}$ release amplitude and kinetics are affected in *Macf1* Cre+ muscle fibers. *Figure 6G* shows rhod-2 $Ca^{2+}$ transients elicited by membrane depolarizing steps of increasing amplitude in a *Macf1* Cre- and in a *Macf1* Cre+ fibers. As routinely observed under these conditions (*Kutchukian et al., 2017*), transients in control fibers exhibit a fast early rising phase upon depolarization followed by a slower phase at low and intermediate voltages and by a slowly decaying phase for the largest depolarizing steps. As shown in *Figure 6H*, rhod-2 transients from *Macf1* Cre+ fibers exhibited an overall similar time-course, not distinguishable from control muscle fibers. In each tested fiber, the rate of SR $Ca^{2+}$ release was calculated from the rhod-2 $Ca^{2+}$ transients. Traces for the calculated rate of SR $Ca^{2+}$ release corresponding to the transients shown in *Figure 6H* are shown in *Figure 6I*. In both fibers, the rate exhibits a similar early peak, whose amplitude increases with that of the pulse, followed by a spontaneous decay down to a low level. The SR $Ca^{2+}$ released peak amplitude was similar in *Macf1* Cre- and in *Macf1* Cre+ muscle fibers for all voltages (*Figure 6I*) and the time to reach the peak was also not affected in the *Macf1* Cre+ muscle fibers (*Figure 6I*, values not shown). Mean values for maximal rate of SR $Ca^{2+}$ release (Max d[$Ca_{tot}$]/dt), mid-activation voltage ($V_{0.5}$) and slope factor (k) of the voltage-dependence were statistically unchanged in *Macf1* Cre+ muscle fibers compared to control fibers (*Figure 6J*). Accordingly, there was also no change in the voltage-dependent $Ca^{2+}$ channel activity of the dihydropyridine receptor (also referred to as $Ca_V1.1$, the voltage-sensor of excitation-contraction coupling) in fibers from *Macf1* Cre+ mice (not shown).

Overall, despite some structure alterations, myofibers isolated from *Macf1* conditional-KO mice exhibit efficient voltage-activated SR-$Ca^{2+}$ release, strongly suggesting that altered force-frequency relationship is related to neuromuscular transmission deficiency.

## MACF1 controls mitochondria biogenesis in skeletal muscle

During our studies on muscle excitability, measurement of the absolute maximal force developed by *Macf1* conditional KO mice was not changed compared to control mice (data not shown) but when we questioned the fatigability of those muscles, we found that in both young and old mice, *Macf1* Cre+ are more resistant to fatigue compared to control mice (*Figure 7A*). Since skeletal muscles are

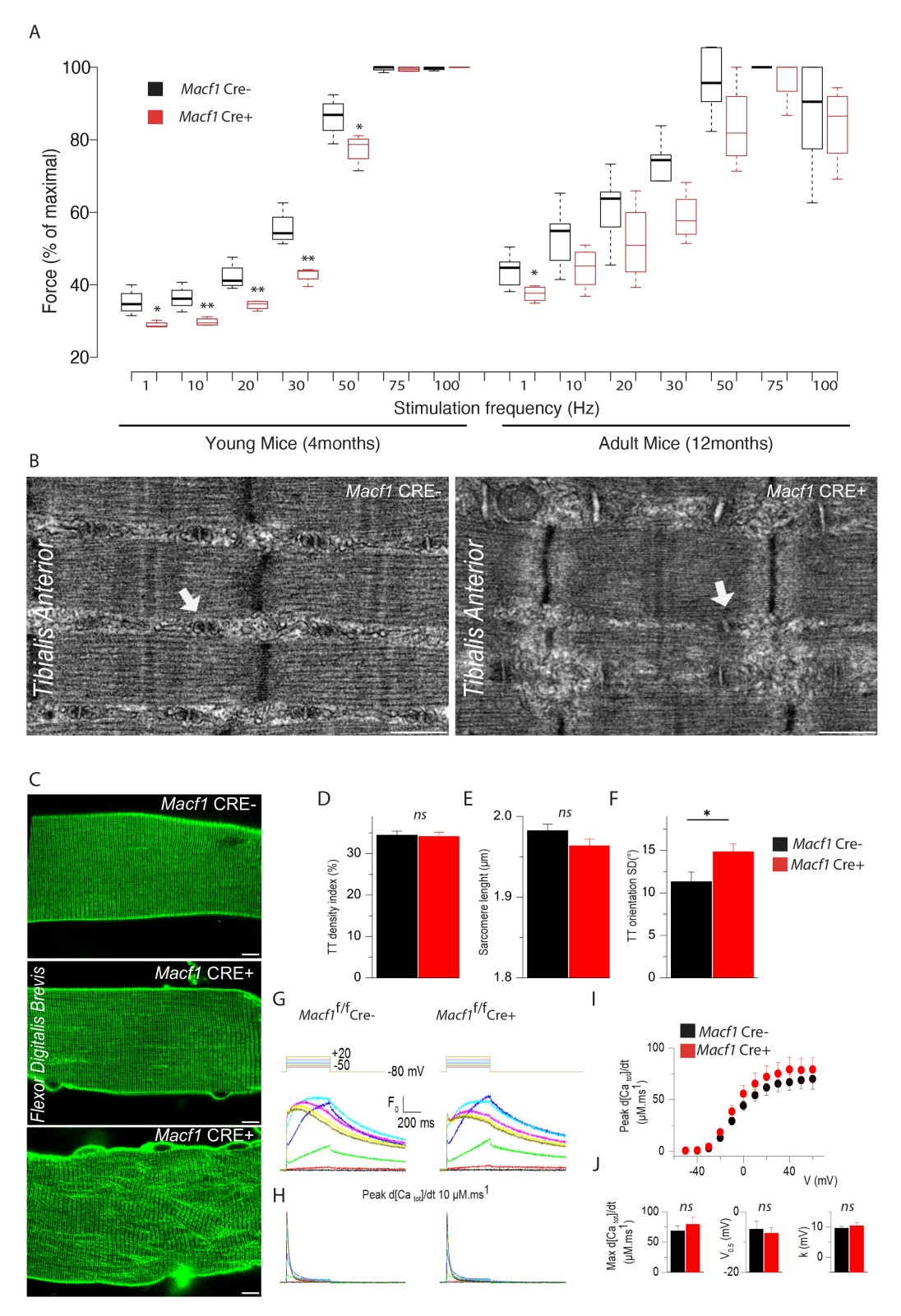

**Figure 6.** Muscle-specific *Macf1* knockout affects muscle force production and T-tubule organization but leaves SR Ca²⁺ release unchanged. (**A**) Force production from the *Hindlimb* muscles in young (4 months) and adult (12 months) *Macf1* Cre- and *Macf1* Cre+ mice in response to various incremental stimulation frequencies (from 1 to 100 Hz). Line is set at median. *p<0.05 and **p<0.01 (Student's t test). (**B**) Representative electron microscopy images of myofibrils and T-tubule organization within *Tibialis Anterior* muscles from 12-month-old *Macf1* Cre- and *Macf1* Cre+ mice. Scale Bar = 0.5 μm. (**C**)
*Figure 6 continued on next page*

*Figure 6 continued*

Representative fluorescence microscopy images of di-8-anepps staining of the T-tubule network in *Flexor Digitalis Brevis* muscle fibers of 12-month-old *Macf1* Cre- and *Macf1* Cre+ mice. Scale Bar = 15 μm. (D–F) Quantification of T-tubules density index (D), Sarcomere length (E) and T-tubules orientation (F) in *Flexor Digitalis Brevis* muscle fibers from 12-month-old *Macf1* Cre- and *Macf1* Cre+ mice. Mean with SD is showing. Ns stands for not significant and *p<0.05 (Student's t test). (G) Representative rhod-2 Ca$^{2+}$ transients in a *Macf1* Cre- and in a *Macf1* Cre+ fiber in response to 0.5 s-long depolarizing pulses from −80 mV to the range of indicated values, with intervals of 10 mV. (H) Corresponding Ca$^{2+}$ release flux (d[Ca$_{Tot}$]/dt) traces calculated as described in the Materials and methods. (I) Mean voltage-dependence of the peak rate of SR Ca$^{2+}$ release in *Macf1* Cre- and in a *Macf1* Cre+ fibers. (J) Inset shows the mean values for maximal rate, half-activation voltage and steepness factor (k) for SR Ca$^{2+}$ release in the two groups of fibers, as assessed from Boltzmann fits to data from each fiber.

The online version of this article includes the following source data for figure 6:

**Source data 1.** Table showing data from experiments plotted in *Figure 6A,D–F and I*.

composed of a functional and metabolic continuum of slow (type I) and fast fibers (types IIa and IIx) known to distinctively rely on metabolic pathways and mitochondrial activity, we first questioned whether there were changes in the mitochondrial pool of adult conditional-KO mice. We measured the intensity of succinate dehydrogenase (SDH) staining, indicative of mitochondrial activity, and found a significant increase in the number of fibers with strong positive succinate dehydrogenase staining in *Tibialis Anterior* muscle from 12-month-old *Macf1* Cre+ compared to control mice (*Figure 7B–C*). As the intensity of SDH staining is commonly used to discriminate slow and fast fibers, we investigated if the proportion of slow fibers was changed in *Macf1* Cre+ mice. Cross-sections of 12-month-adult muscles from the *Tibialis Anterior, Soleus* and *Gastrocnemius* were analyzed for slow myosin content and no alteration was observed (*Figure 7—figure supplement 1A–B*).

We next quantified the content of mitochondria and of proteins of the electron transport chain. There was an increase in the total amount of mitochondria (Tom20 relative to actin) in total protein extracts from *Gastrocnemius* of 12-month-old *Macf1* Cre+ mice, as compared to control mice. However, there were no changes in the levels of mitochondrial electron transport chain proteins (CI, II, III, IV, and V, relative to Tom20) (*Figure 7D*). To confirm the increase of mitochondria pool, we measured the ratio of Mitochondrial-DNA to Genomic-DNA. This ratio did not differ between young (3-month-old) and adult (12-month-old) control mice, nor did it between young *Macf1* Cre+ and control mice (*Figure 7E*). Interestingly, adult *Macf1* Cre+ mice showed an increased amount of mitochondrial DNA compared to the other groups (*Figure 7E*). To understand this phenomenon, we followed the expression level of different genes known to contribute to mitochondria fusion/fission/biogenesis such as *DNM1, DNM2, PPARGC1A* and *SLN.* Remarkably, we found that only the expression of Sarcolipin, known to promote mitochondria biogenesis in skeletal muscle (*Maurya et al., 2018*; *Maurya et al., 2015*) was increased in *Macf1* Cre+ mice compared to control mice (*Figure 7F*, *Figure 7—figure supplement 1C*). We next checked the staining of Cytochrome C, indicative of mitochondria content and spatial organization, in myofibers isolated from *Tibialis Anterior* muscles from 12-month-old mice. Our immunofluorescence approach showed a strong accumulation of mitochondria in different areas at the periphery of myofibers from the conditional-KO mice compared to control mice (*Figure 7G*, arrow). Additionally, there was an accumulation of mis-oriented longitudinal staining (*Figure 7H*), reminiscent of what was observed with the T-tubules staining in certain *Macf1* Cre+ myofibers (*Figure 6C*). This suggests an alteration of myofibrils cohesion at the periphery of the myofibers. In this view, since mitochondria are linked to both the microtubule network and to Desmin intermediate filaments (*Reipert et al., 1999*), we questioned the integrity of the Desmin network and found a similar disruption as observed for mitochondria staining (*Figure 7—figure supplement 1D-E*). Electron microscopy was next used to visualize the ultrastructure of mitochondria in *Tibialis Anterior* muscle of *Macf1* Cre+ mice. This approach confirmed the increase in mitochondria content in *Macf1* Cre+ myofibers, associated with the presence of spherical mitochondria in-between myofibrils (*Figure 7I*). We next questioned if the mitochondrial network distribution was altered in vitro, in primary myofibers (*Figure 7J*). In myofibers treated with scrambled siRNA/shRNA, mitochondria lined up in the bulk of myofibers, in-between myofibrils, where they followed the longitudinal axis, just like the microtubule network. However, in *Macf1* siRNA/shRNA-treated myofibers, we observed a 30% increase of mitochondria content (*Figure 7K*). These results confirmed a role of MACF1 in muscles specifically related to mitochondria biogenesis. Consequently, MACF1 is involved

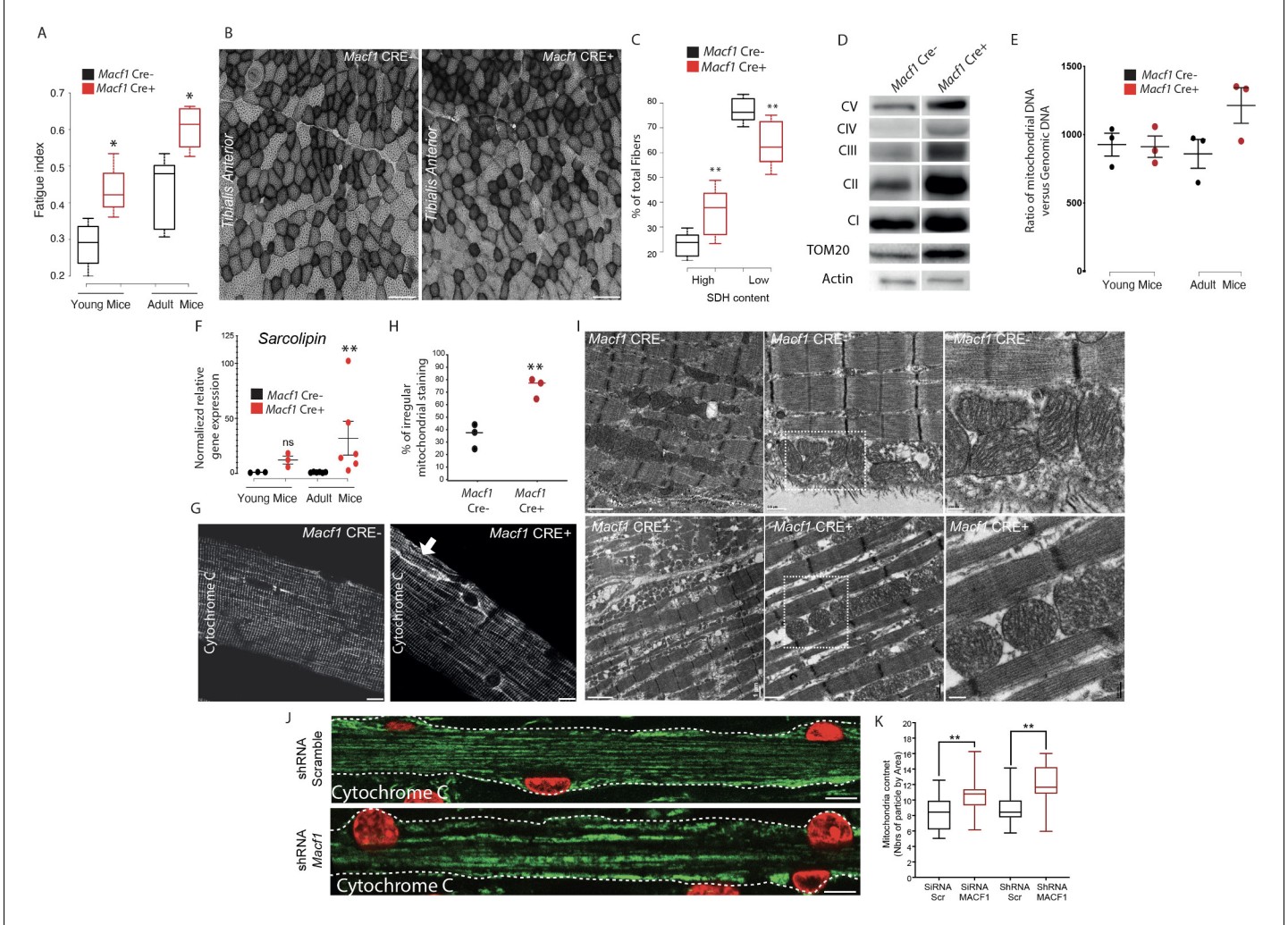

**Figure 7.** Muscle specific MACF1 knockout affects mitochondrial content and organization. (**A**) Quantification of the fatigue index from the *Hindlimb* muscles in young (4 months) and adult (12 months) *Macf1* Cre- and *Macf1* Cre+. Line is set at median. *p<0.05 (Student's t test). (**B**) Representative images of transversal cross-section of *Tibialis Anterior* muscle from 12-month-old *Macf1* Cre- (left panel) and *Macf1* Cre+ (right panel) mice stained for Succinate DeHydrogenase activity. Scale Bar = 150 μm. (**C**) Quantification of the Succinate DeHydrogenase activity relative to myofibers distribution in conditional *Macf1*-KO mice compared to control mice. Line is set at median. Ns stands for not significant, **p<0.01 (Student's t test). (**D**) Western blot analysis of OXPHOS mitochondrial electron transport chain, TOM20 and actin proteins expression in total extracts from *Gastrocnemius* muscles of 12-month-old *Macf1* Cre- and *Macf1* Cre+ mice. (**E**) QPCR quantification of the ratio between mitochondrial and genomic DNA in total DNA extracts obtained from *Gastrocnemius* muscles of three individual *Macf1* Cre- and *Macf1* Cre+ mice at each given age. Line is set at mean (with SEM). (**F**) QRT-PCR results presenting relative gene expression level for SLN in *Macf1* Cre- and Cre+ at each given age. Following the calculation of relative gene expression level with reference to housekeeping genes for each mouse, the mean expression level of SLN was calculated for the WT mice. Next, the expression level of each WT and KO mouse was normalized to the previously calculated mean value. Line is set at mean (with SEM). Ns stands for not significant and **p<0.01 (Mann-Whitney test). (**G**) Representative confocal images (63×) from isolated myofibers of *Tibialis Anterior* stained for Cytochrome C in *Macf1* Cre- (left panel) and Cre+ (right panel) mice of 12-month-old mice. Scale Bar = 15 μm. (**H**) Quantification of the percentage of myofibers with irregular (broken/oriented/accumulated) mitochondrial staining in isolated myofibers of *Tibialis Anterior* muscles from three individual *Macf1* Cre- and Cre+ mice at the age of 12 months. Line is set at median. **p<0.01 (Student's t test). (**I**) Representative electron microscopy images of myofibrils and mitochondria organization within *Tibialis Anterior* muscles from 12-month-old *Macf1* Cre- (upper panels) and *Macf1* Cre+ (lower panels) mice. Scale Bar = 1, 0.5, 0.2 μm. (**J**) Confocal immunofluorescence images (63×) presenting the organization of Cytochrome C (green) in primary myofibers treated either with scramble shRNA (upper panel) or with a pool of four distinct shRNAs targeting *Macf1* (lower panel) after 13 days of differentiation. Myonuclei are presented in red. Scale Bar = 10 μm. (**K**) Quantification of mitochondrial content per area in primary myofibers treated either with scramble shRNA or with a pool of four distinct shRNAs targeting *Macf1* after 13 days of differentiation. Line is set at median. **p<0.01 (Student's t test).

The online version of this article includes the following source data and figure supplement(s) for figure 7:

**Source data 1.** Table showing data from experiments plotted in *Figure 7A,C,E–F,H and K*.

*Figure 7 continued on next page*

*Figure 7 continued*

**Source data 2.** Whole immunoblotting membrane for OXPHOS, TOM20, and actin plotted in *Figure 7D*.
**Figure supplement 1.** MACF1 conditional KO mice present normal distribution of different skeletal muscles types but disorganized Desmin network.
**Figure supplement 1—source data 1.** Table showing data from experiments plotted in *Figure 7—figure supplement 1B–C and E*.

in the regulation of mitochondria biogenesis as loss of MACF1 is associated with an increase in Sarcolipin level and redistribution of mitochondrial content.

## Discussion

Different mechanisms involving actin and microtubule network control myonuclei positioning during muscle fiber development and differentiation (*Azevedo and Baylies, 2020*). However, the long-term mechanisms allowing myonuclei patterning maintenance in mature muscle fibers are still pending. The present study describes a MACF1-dependent pathway that is critical for proper myonuclei distribution at the periphery of the myofibers, through the ability of MACF1 to control microtubule network dynamics. MACF1 is a giant member of the spectraplakin family, able to simultaneously bind to different cytoskeleton members. Spectraplakins are involved in intracellular trafficking, cellular polarization migration and shaping, cell-cell adhesion and mechanical strength (*Suozzi et al., 2012*). In mononuclear cells, MACF1 acts as a hub between the actin and microtubule networks to promote their alignment, allowing cells structure maintenance (*Preciado López et al., 2014*). In skeletal muscle, MACF1 has been implicated in the integrity of the NMJ through its association with rapsyn and actin (*Antolik et al., 2007*; *Oury et al., 2019*). Shot, the *Drosophila* orthologue participates in the formation and maintenance of a perinuclear shield in muscle (*Wang et al., 2015*). Still, MACF1 function in muscle remained puzzling as *drosophila* Shot exhibits a myonuclei-related localization while previous studies in murine models identified MACF1 specifically accumulated at the NMJ.

Our in vitro results show that MACF1 is progressively expressed during muscle fiber maturation and differentiation and accumulates at the periphery of the myofibers where it colocalizes with cortical microtubules (*Figure 1*, *Figure 1—figure supplement 1*). In vivo, muscle MACF1 also accumulates strongly in the vicinity of extra-synaptic myonuclei and colocalizes mainly with the longitudinal microtubule network (*Figure 3*), in accordance with observations from *Drosophila* myofibers (*Figure 5—figure supplement 1*; *Wang et al., 2015*). Interestingly, the MACF1 accumulation pattern along the extrasynaptic myofibers differs from that at the NMJ where there isn't a particular MACF1 meshwork around synaptic myonuclei, nor a colocalization of MACF1 with AChRs (*Figure 5*, *Figure 5—figure supplement 1*). Accordingly, extra-synaptic myonuclei positioning is altered in developing myofibers of MACF1-depleted muscles (*Figures 1* and *4*), whereas synaptic myonuclei positioning is not affected (*Figure 5*). Our in vitro results suggest that the progressive alteration of myonuclei positioning is mainly due to the capacity of MACF1 to freeze myonuclei displacement events (*Figure 2*). Thus, in the MACF1-depleted condition, myonuclei motion is increased and contributes to the alteration of myonuclei distribution along myofibers that translates into a reduction in the distance between adjacent myonuclei (*Figures 1* and *4*). Interestingly, the in vivo long-term depletion of muscle MACF1 leads to a dramatic change in the location of myonuclei in 12 month-old mice. In these mice, myonuclei are internalized, with no associated signs of regeneration (*Figure 4*, *Figure 4—figure supplement 1*). Myonuclei internalizations is concomitant with the appearance of T-tubules enlargement and with continuity breaks of membranes and Desmin structures in the myofibers (*Figures 6* and *7*). In mature myofibers, myonuclei have been found to reach the periphery of myofibers by a process dependent on an interplay between myonuclei rigidity, actin and Desmin network (*Roman et al., 2017*). Depletion of MACF1 in vitro or in vivo does not impact the actin network integrity, as reflected by unmodified sarcomere's structure (*Figures 1* and *3*, *Figure 3—figure supplement 1*). In contrast, the myonuclei morphology is modified (*Figures 1* and *4*). We believe that in MACF1 depleted myofibers, the loosened anchorage of peripheral myonuclei is the initial cause of changes in their morphology. Over time, MACF1 depletion affects the Desmin network, which is a myofibril linker known to play a hub role between the sarcolemmal cytoskeleton and nuclei (*Hnia et al., 2015*). In this perspective, myonuclei internalization in MACF1-depleted myofibers from old mice is likely the consequence of loosening of the MACF1-dependent meshwork in the vicinity

of the myonuclei. This contributes to the long-term enhancement of myonuclei motion, associated with myonuclei shape alteration. Altogether, this internalization requires loss of cohesion within myofibrils, achieved through reorganization of the Desmin network.

Our study shows that MACF1 controls myonuclei displacement through the regulation of microtubules dynamics during myofibers maturation (*Figures 2* and *3*). Early steps of myonuclei positioning in immature myofibers (myotubes) depend on an interplay between microtubules, microtubule-associated-proteins and motors proteins such as MAP4, MAP7, Dynein, and Kif5b (*Cadot et al., 2012*; *Gache et al., 2017*; *Metzger et al., 2012*; *Mogessie et al., 2015*). Interestingly, we found that MACF1 is not involved in the early steps of myonuclei spreading in myotubes (*Figure 1—figure supplement 1*). Using EB3 comets tracking, we show that microtubule dynamic progressively decreases within myofibers during maturation, leading to more stabilized microtubules structures (*Figure 2*). Interestingly, the absence of MACF1 impairs this stabilization and leads to microtubule with a higher polymerization rate (*Figure 2*). We propose that this enhanced dynamic is correlated with the increased myonuclei movement along myofibers, especially because endogenous MACF1 is in vitro and in vivo clearly located around myonuclei and is co-localized with the longitudinal microtubule network (*Figures 1* and *3*). Progressive microtubule network stabilization during myofibers maturation is associated with de-tyrosinated tubulin accumulation. Interestingly, our microscopy images show that a de-tyrosinated pool of microtubules is present in the vicinity of myonuclei in control mice (*Figure 3*; *Gadadhar et al., 2017*). In accordance with the elevated microtubule dynamics, de-tyrosinated tubulin staining around myonuclei is lost in *Macf1* KO muscles (*Figure 3*), supporting a link between microtubule dynamics and myonuclei motion. No direct interplay between MACF1 and tubulin post translational modification (PTMs) has been so far documented, but one can hypothesize that regarding already described MACF1 localization, at the Minus-End of microtubules throught CAMSAP/patronin or at the Plus-End of microtubules throught EB1, MACF1 can influence localization of enzymes controlling tubulin PTMs such as TTL or SVBP (*Aillaud et al., 2017*; *Noordstra et al., 2016*; *Peris et al., 2006*; *Su et al., 2020*). Thus, it is tempting to postulate that in MACF1-depleted myofibers from old mice, this elevated microtubule dynamic is constantly challenging the cohesion between myofibrils through Desmin reorganization, hence contributing to the myonuclei internalization process (*Heffler et al., 2020*). Interestingly, in the MACF1-depleted myofibers, the tyrosinated/de-tyrosinated ratio of microtubules is not changed at the NMJ supporting alternative tubulin PTMs in the maintenance of the microtubule network cohesion at that location (*Osseni et al., 2020*). Nonetheless, we show that MACF1 is important for the control of AChR clustering in mature myofibers (*Figure 5*). Failure of this mechanism in the MACF1-depleted myofibers is the most likely reason for the observed decrease in muscle excitability since it will affect neuromuscular transmission (*Figure 6*). This interpretation is strengthened by the fact that myofibril genesis is not impacted neither in vitro nor in vivo upon MACF1-depletion and by the fact that there is also no alteration in SR-$Ca^{2+}$ release even though the overall structure of T-tubule network was found affected in some of the myofibers (*Figure 6*).

Our study has also revealed an unexpected role for muscle MACF1 related to mitochondria biogenesis. We observed an increase in the number of oxidative fibers in MACF1-depleted muscles from old mice, with no associated change in the myosins expression levels (*Figure 7*, *Figure 7—figure supplement 1*). This alteration is correlated with an increase of mitochondrial DNA and with the increased fatigue resistance of the MACF1 KO mice (*Figure 7*). In addition, our electron and fluorescence microscopy images revealed an alteration of mitochondria shape and patterning as they appear rounder and accumulated in-between myofibrils (*Figure 7*). Molecular motors such as Kif5b and Dynein control mitochondrial motion in myoblasts (iqbal and *Iqbal and Hood, 2014*). Microtubule network disorganization and loss of myofibrils cohesion may explain mitochondria mislocalization along myofibers. Accordingly, MACF1 has been previously implicated in mitochondria localization and nuclear positioning in oocytes (*Escobar-Aguirre et al., 2017*). In addition, an increase in microtubules instability directly impacts the local concentration of free tubulin (*Gache et al., 2005*) and free tubulin-βII level can affect mitochondrial metabolism and organization (*Guerrero et al., 2010*; *Guzun et al., 2011*; *Kumazawa et al., 2014*). Finally, our investigations revealed that the mRNA levels of proteins related to mitochondrial fusion/fission such as DNM1 and DNM2 were unchanged in MACF1 depleted myofibers but the transcripts of Sarcolipin, a protein related to mitochondrial biogenesis, were dramatically increased in conditional old mutant mice compared to controls. One possibility is that this alteration is due to the aberrant myonuclei size/

shape of MACF1 KO myofibers since it has been reported that nuclear alterations can impact gene expression through changes in mechanotransduction processes (*Robson et al., 2016*; *Wang et al., 2018*). In any case, our results indicate that the increase in mitochondria number is already present in young mice (*Figure 7*) and is the most likely explanation for the resistance of both young and old KO mice to fatigue. We thus provide evidence that microtubule organization and dynamics in muscle cells are imporant not only for mitochondria motion and positioning, but also to maintain mitochondria biogenesis (*Mado et al., 2019*).

The present results have physiopathological relevance as there are several disease conditions associated with peripheral myonuclei mis-localization in muscle fibers. This is the case for a heterogeneous group of inherited muscle diseases, called CentroNuclear Myopathies (CNMs), which are defined pathologically by an abnormal localization of myonuclei in the center of the myofibers (*Jungbluth and Gautel, 2014*). Genes implicated in the various forms of CNMs encode proteins that participate in different aspects of membrane remodeling and trafficking such as MTM1, Amphyphisin-2/BIN1, DNM2, and HACD1 (*Bitoun et al., 2005*; *Laporte et al., 1996*; *Muhammad et al., 2013*; *Muller et al., 2003*). Another example is skeletal muscle biopsies from patients with desminopathies which also present internalized myonuclei (*Clemen et al., 2013*). Here, we have identified MACF1, as a critical contributor in the maintenance of extra-synaptic myonuclei positioning, in the maintenance of the NMJ integrity and in mitochondria location and biogenesis. This is of strong relevance to our understanding of muscle diseases associated with centralization of myonuclei and potentially to other pathological muscle defects.

# Materials and methods

## Cell culture

Primary myoblasts were collected from wild type C57BL6 mice as described before (*Falcone et al., 2014*; *Pimentel et al., 2017*). Briefly, Hindlimb muscles from 6 days pups were extracted and digested with collagenase (Sigma, C9263-1G) and dispase (Roche, 04942078001). After a pre-plating step to discard contaminant cells such as fibroblasts, myoblasts were cultured on 1% matrigel coated-dish (Corning, 356231) in growth media (GM: IMDM (Gibco, 21980–032), 20% fetal bovine serum (Gibco, 10270–106), 1% chicken embryo extract (USBiological, C3999), and 1% penicillin-streptomycin (Gibco, 15140–122)). When reached to 90% of confluence, myocytes were induced to differentiate in myotubes for 2–3 days in differentiation media (DM: IMDM (Gibco, 21980–032), 2% of horse serum (Gibco, 16050–122) and 1% penicillin-streptomycin (Gibco, 15140–122)). Myotubes were then covered by a concentrated layer of matrigel and maintained for up to 10 days in long differentiation culture medium (LDM: IMDM (Gibco, 21980–032), 2% of horse serum (Gibco, 16050–122), 0.1% Agrin + 1% penicillin-streptomycin (Gibco, 15140–122)) until the formation of mature and contracting myofibers. LDM was changed every two days.

Mouse myoblast C2C12 cells (ATCC, CRL-1772, RRID:CVCL_0188), tested negative form mycoplasma, were cultured in growth media (DMEM (Gibco, 41966029), 15% fetal bovine serum (Gibco, 10270–106) and 1% penicillin-streptomycin (Gibco, 15140–122)) and were plated on 1% Matrigel-coated dishes for 1–2 days before differentiation. Differentiation was induced by switching to differentiation media (DMEM (Gibco, 41966029)) with 1% horse serum (Gibco, 16050–122).

## Production of wild-type Agrin recombinant proteins

For the production of recombinant proteins, the stably transfected HEK293-EBNA cells were grown to about 80% confluence and were transferred to expression medium without FBS. Conditioned medium containing secreted proteins was collected every 3 days and replaced with fresh expression media for 12 days. Conditioned medium was centrifuged at 2000xg for 10 min to pellet the cells before storing at −20°C. After thawing, recombinant proteins were purified from conditioned media by HPLC, on a HiTRAP lmac HP column (GE Healthcare, 17-0920-03), eluted with imidazol and desalted on silica column with PBS (GE Healthcare, HiPrep 26 /10Desalting) or on a Vivaspin column (Vivaspin Sartorius). The absolute concentration of soluble Agrins was estimated on a coomassie blue stained gel by comparison with a known amount of a commercial purified Agrin.

## Cell transfection

For C2C12 cells, scramble or *Macf1* targeting siRNAs were transfected in cells using Lipofectamine 2000 (ThermoFisher Scientifics, 11668–019) at the final concentration of 10 nM, following manufacturer instructions, just before differentiation. Same transfection methode was used according to the instructions to deliver scramble or *Macf1* targeting shRNAs to cells.

For primaries cells, siRNAs were transfected using Lipofectamine 2000 (ThermoFisher Scientifics, 11668–019) at the final concentration of 2 nM. shRNAs (Geneocopia), MACF1-MTBD-GFP (gift from Carsten Janke team, Institut Curie, Paris, France), EB3-GFP (gift from Annie Andrieux team, Grenoble Institute Neurosciences, Grenoble, France) or RFP-Lamin-chromobody (Chromotek) cDNA were transfected to cells using Lipofectamine 3000 (ThermoFisher Scientifics, L3000-008) following manufacturer instructions, just before differentiation.

| siRNA | Sense oligonucleotide sequence | Anti-Sense oligonucleotide sequence |
|---|---|---|
| MACF1 #952 | GAGUUUCAAAAGAACCUUAtt | UAAGGUUCUUUUGAAACUCtt |
| MACF1 #954 | CAAUACAGCUGAAAAAGUUtt | AACUUUUUCAGCUGUAUUGgg |
| MACF1 #958 | GGAUGAAAUUAAUACUCGAtt | UCGAGUAUUAAUUUCAUCCat |
| shRNA | Clone Name | Target Sequence |
| MACF1 #1 | MSH071923-1-CU6 (OS398499) | GCACATCAATGATCTCTATGA |
| MACF1 #2 | MSH071923-2-CU6 (OS398500) | GCAGGTGAAGCTAGTGAATAT |
| MACF1 #3 | MSH071923-3-CU6 (OS398501) | GCAGATTGCAAACAAGATACA |
| MACF1 #4 | MSH071923-4-CU6 (OS398502) | GCTAAAGAATATCCGACTACT |

## Mouse model

Mice that carry a loxP-flanked allele of *Macf1* (*Goryunov et al., 2010*) and Hsa-Cre transgenic mice (*Miniou, 1999*) have been described previously. All of the experiments and procedures were conducted in accordance with the guidelines of the local animal ethics committee of the University Claude Bernard – Lyon one and in accordance with French and European legislation on animal experimentation and approved by the ethics committee CECCAPP and the French ministry of research (CECCAPP_ENS_2018_022).

## In vivo force measurements

Mice were initially anesthetized in an induction chamber using 4% isoflurane. The right *Hindlimb* was shaved before an electrode cream was applied at the knee and heel regions to optimize electrical stimulation. Each anesthetized mouse was placed supine in a cradle allowing for a strict standardization of the animal positioning. Throughout a typical experiment, anesthesia was maintained by air inhalation through a facemask continuously supplied with 1.5% isoflurane. The cradle also includes an electrical heating blanket in order to maintain the animal at a physiological temperature during anesthesia. Electrical stimuli were delivered through two electrodes located below the knee and the Achille's tendon. The right foot was positioned and firmly immobilized through a rigid slipper on a pedal of an ergometer (NIMPHEA_Research, AII Biomedical SAS) allowing for the measurement of the force produced by the Hindlimb muscles (i.e. mainly the *Gastrocnemius* muscle). The right knee was also firmly maintained using a rigid fixation in order to optimize isometric force recordings. Monophasic rectangular pulses of 0.2 ms were delivered using a constant-current stimulator (Digitimer DS7AH, maximal voltage: 400V). The force-frequency curves were determined by stepwise increasing stimulation frequency, with resting periods > 30 s between stimuli in order to avoid effects due to fatigue. For each stimulation train, isometric peak force was calculated. After a 3 min recovery period, force was assessed during a fatigue protocol consisting of 30 Hz stimulation trains of 0.3 s delivered once every second for 180 s. The peak force of each contraction was measured and averaged every five contractions. A fatigue index corresponding to the ratio between the last five and the first five contractions was determined. Force signal was sampled at 1000 Hz using a Powerlab system and Labchart software (ADinstruments).

## Sciatic nerve crush

The mice were made unconscious by isoflurane anesthesia prior to the intraperitoneal injection of ketamine (100 mg/kg) and xylazine (10 mg/kg) in order to obtain a deep state of general anesthesia. To limit the pain on awakening, mice received a subcutaneous injection of buprenorphine at 30 µg/ml (0.1 µg/g of bodyweight). Once deeply anesthetized mice were shaved at the level of the left hip and the skin incised over 3 mm, then the connective tissue (muscle fascia) was incised. In order to reach the sciatic nerve the muscle mass was pushed back with the delicate help of fine forceps. The exposed sciatic nerve was subjected for two minutes to a drop of lurocaïne in order to inhibit any nociceptive impulse. Then the nerve was seized using fine forceps and sectioned by ablation of 2–3 mm to avoid any reconnection by scarring. The wound was then stitched up using suture. 15 days later, after killing the mice according to the ethical rules of the establishment, *Tibialis anterior* muscles were collected for examination.

## Mouse genotyping

In order to extract the DNA, samples (mice tales) were incubated in extraction solution (25 mM NAOH, 0.2 mM EDTA) for 30 min at 95℃. Following the addition of neutralization solution (40 mM Tris-HCL), samples were centrifuged at 13000 RPM for 2 min at RT and the DNA were collected. DNA concentration was evaluated using Nanodrop (ThermoFisher Scientifics). PCR were carried (Hot Start Taq polymerase (QIAGEN, 1007837), PCR buffer mix (QIAGEN, 1005479) DNTP mix (Biolabs, 447)) using the following primers. The same amount of PCR products was loaded in 2X Agarose-1X TAE gels and were migrated for 2H at 130V. Results were obtained using GelDoc (BioRad).

List of the primers used for PCR:

| Primer | Sequence |
| --- | --- |
| MACF1 F | CATCAGAAGAGATCAACCAACC |
| MACF1 R | AAAGGAAGAGAGGTCCAAGGT |
| Cre F | GAGTTGATAGCTGGCTGGTGGCAGATG |
| Cre R | CCTGGAAAATGCTTCTGTCCGTTTGCC |
| CD8 F | GGTGCATTCTCACTCTGAGTTCC |
| CD8 R | GCAGACAGAGCTGATTTCCTATGTG |

## Protein extraction, western blot, and dot blot analysis

For primary cultured cells or C2C12 cell lines, cells were harvested, using 1X Trypsin for 5 min at 37℃ and centrifuged at 1500 RPM for 5 min at 4℃. Cell pellets were diluted and incubated in the optimal volume of RIPA lysis buffer containing phosphatases inhibitors (Sigma, P5726-5mL) and proteases inhibitors (Sigma, P8340) for 30 min at 4℃. Following a sonication and a centrifugation at 12,000 RPM for 10 min at 4℃, protein samples were collected for further uses. The concentration of proteins was determined using BCA protein assay kit (Thermo Fisher Scientifics, 23225) as described by the manufacturer.

To obtain protein extracts from dissected muscles or other organs, the samples were finely cut onto tubes containing ceramic beads (MP Biomedicals Lysing Matrix D, 6913100) with the optimal volume of RIPA lysis buffer completed with phosphatases inhibitors (Sigma, P5726-5mL) and proteases inhibitors (Sigma, P8340). Tubes were subjected to harsh shaking (6500 rpm, 3 cycles of 15 s with 30 s pause intervals at 4℃) and rested at 4℃ for 1H. Following a sonication and a centrifugation at 12000 RPM for 10 min at 4℃, protein samples were collected for further uses. The concentration of proteins was determined using BCA protein assay kit (Thermo Fisher Scientifics, 23225) as described by the manufacturer.

To carry out dot blots, Bio-Dot SF Microfiltration Apparatus plates (BioRad) were used to transfer the protein samples onto the nitrocellulose membranes following protein extraction and dosage.

Western blot or dot blot Membranes were then saturated in 5% milk in TBS for 1H at RT and were incubated in primary antibodies over night at 4℃. Following washes by 0.1% Tween-20-1X TBS, the membranes were incubated in HRP conjugated secondary antibodies in 5% milk in TBST for

1H at RT. Following washes by 0.1% Tween-20 in TBS the detection of the target proteins was carried out using Super Signal West Femto (Thermo Fisher Scientifics, 34095) and ChemiDoc imaging system (BioRad).

List of antibodies used for western blot:

| Antibody | Species and utility | Dilution factor | Manufacturer and reference |
|---|---|---|---|
| Anti-MACF1 | Rabbit - primary | 1:1000 | Novus Biologicals NBP2-36528 (RRID:AB_2893083) |
| Anti-GAPDH | Mouse - primary | 1/1000 | Sigma-Aldrich MAB374 (RRID:AB_2107445) |
| Anti-TOM20 | Rabbit - primary | 1:1000 | Cell Signaling Technologies D8T4N (RRID:AB_2687663) |
| Anti-OXPHOS | Mouse - primary | 1:1000 | Abcam Ab110413 (RRID:AB_2629281) |
| Anti-mouse-HRP | Goat - secondary | 1:5000 | Invitrogen 62–6520 (RRID:AB_2533947) |
| Anti-rabbit-HRP | Goat - secondary | 1:5000 | Invitrogen 65–6120 (RRID:AB_2533967) |

## Primary cells immunofluorescence staining

Cells were fixed in 4% PFA in PBS for 20 min at 37°C followed by washes with PBS and permeabilization with 0.5% Triton-X100 in PBS for 5 min at RT. Following washes with PBS, cells were saturated with 1% BSA in PBS for 30 min at RT and incubated in primary antibodies over night at 4°C. After several washes with 0.05% Triton-X100-1X PBS, cells were incubated in secondary antibodies or dyes for 2H at RT and washed with 0.05% Triton-X100 in PBS before image acquisition. The list of the antibodies used for each staining can be found in the following section.

## Isolation of mono-myofibers and immunofluorescence staining

Following the dissection of the whole muscle from the mice, muscle blocks were fixed in 4% PFA in PBS for 2H at RT. After several washes, 30– 50 mono-myofibers were isolated per staining from each muscle. Myofibers were then permeabilized using 1% Triton-X100 in PBS for 15 min at 37°C and saturated in 1% BSA in PBS for 30 min at RT. Based on the experiments, myofibers were incubated in desired primary antibodies at 4°C for two nights. Following washes with 0.05% Triton-X100 in PBS, myofibers were incubated in secondary antibodies or dyes for 2H at RT and washed several times with 0.05% Triton-X100 in PBS before mounting on slides. Myofibers were mounted on slides using fluromount Aqueous mounting (Sigma, F4680-25mL) and kept at 4°C utile image acquisition.

List of the antibodies used for immunofluorescent staining:

| Antibody or dye | Species and utility | Dilution factor | Manufacturer and reference |
|---|---|---|---|
| Anti-Cytochrome C | Mouse - primary | 1:300 | Cell Signaling Technologies 12963S (RRID:AB_2637072) |
| Anti-MACF1 | Rabbit - primary | 1:200 | Novus Biologicals NBP2-36528 (RRID:AB_2893083) |
| Anti-alpha tubulin | Mouse - primary | 1:1000 | Sigma T6074 (RRID:AB_477582) |
| Anti-Tyrosinated tubulin | Rat - primary | 1:100 | Andrieux's Lab gift |
| Anti-De-tyrosinated tubulin | Rabbit - primary | 1:100 | Andrieux's Lab gift |
| MF20 | Mouse-primary | 1:10 | DSHB MF20 (RRID:AB_2147781) |
| Anti-alpha actinin | Mouse – primary | 1:350 | Sigma A7811 (RRID:AB_476766) |

*Continued on next page*

*Continued*

| Antibody or dye | Species and utility | Dilution factor | Manufacturer and reference |
|---|---|---|---|
| Anti-Desmin | Rabbit - primary | 1:500 | Cell Signaling Technologies D93F5 (RRID:AB_1903947) |
| Dapi-brilliant blue | Dye | 1:50000 | Thermo Fisher Scientifics D1306 (RRID:AB_2629482) |
| α-Bungarotoxin-Alexa Flour 488 | Dye | 1:1000 | Thermo Fisher Scientifics B13422 |
| α-Bungarotoxin-Alexa Flour 555 | Dye | 1:200 | Thermo Fisher Scientifics B35451 |
| Phalloidin-Alexa Flour 647 | Dye | 1:100 | Thermo Fisher Scientifics A22287 |
| Anti-rat-Alexa Flour 488 | Donkey - secondary | 1:500 | Thermo Fisher Scientifics A-21208 (RRID:AB_2535794) |
| Anti-rabbit-Alexa Flour 647 | Goat - secondary | 1:500 | Thermo Fisher Scientifics A-21245 (RRID:AB_2535813) |
| Anti-mouse-Alexa Flour 647 | Goat - secondary | 1:500 | Thermo Fisher Scientifics A-21240 (RRID:AB_2535809) |

## Histological staining and analysis

Tibialis anterior, *Soleus* and *Gastrocnimius* muscles were collected, embedded in tragacanth gum, and quickly frozen in isopentane cooled in liquid nitrogen. Cross-sections were obtained from the middle portion of frozen muscles and processed for histological, immunohistochemical or enzymo-histological analysis according to standard protocols.

The fiber cross-sectional area and the number of centrally nucleated fibers were determined using Laminin (Sigma L9393) and Dapi-stained sections. Slow myosin antibody (MyHC-I BA-D5) was used to classify muscle fibers types.

SDH staining was performed as previously described (*Nachlas et al., 1957*). Briefly, transverse sections (8 µm) were cut from the mid-belly of the TA muscles on a cryostat at −20°C and stored at −80°C until SDH staining was performed. The sections were dried at room temperature for 30 min before incubation in a solution made up of 0.2M phosphate buffer (pH 7.4), 0.1M $MgCl_2$, 0.2M Succinic Acid (Sigma), and 2.4 mM NitroBlue Tetrazolium (NBT, Sigma) at 37°C in a humidity chamber for 45 min. The sections were then washed in deionized water for 3 min, dehydrated in 50% ethanol for 2 min, and mounted for viewing with DPX mount medium (Electron Microscopy Sciences). Images were acquired as described above.

Fluorescence microscopy and transmission microscopy were performed using Axioimager Z1 microscope with CP Achromat 5x/0.12, 10x/0.3 Ph1, or 20x/0.5 Plan NeoFluar objectives (Zeiss). Images were captured using a charge- coupled device monochrome camera (Coolsnap HQ, Photometrics) or color camera (Coolsnap colour) and MetaMorph software. For all imaging, exposure settings were identical between compared samples. Fiber number and size, central nuclei and peripheral myonuclei were calculated using ImageJ software.

## Analysis of T-tubule network and intracellular $Ca^{2+}$ in voltage-clamped fibers

*FDB* muscle fibers were incubated for 30 min in the presence of 10 µm di-8-anepps in Tyrode solution. Estimation of the T-tubule density from the di-8-anepps fluorescence was carried out from a largest possible region of interest excluding the plasma membrane, within each fiber. For each fiber, two images taken at distinct locations were used. Analysis was carried out with the ImageJ software (National Institute of Health). Automatic threshold with the Otsu method was used to create a binary image of the surface area occupied by T-tubules. The 'skeletonize' function was then used to delineate the T-tubule network. T-tubule density was expressed as the percent of positive pixels within the region. Sarcomere length was estimated from half the number of fluorescence peaks (T-tubules) along the length of the main axis of a given fiber. To assess variability in T-tubule orientation, objects within two T-tubule binary images of each fiber were outlined and particle analysis was performed to

determine the angle of all objects yielding a perimeter larger than an arbitrary value of 10 µm. For each fiber, the standard deviation of angle values was then calculated. This analysis was performed on 10 muscle fibers from 3 Macf1f/f Cre- and from 3 Macf1f/f Cre+ mice, respectively.

Single fibers were isolated from *FDB* muscles as described previously (Jacquemond, 1997). In brief, muscles were incubated for 60 min at 37°C in the presence of external Tyrode containing 2 mg.mL collagenase (Sigma, type 1). Single fibers were obtained by triturating the collagenase-treated muscles within the experimental chamber.

Isolated muscle fibers were handled with the silicone voltage-clamp technique (*Lefebvre et al., 2014*). Briefly, fibers were partly insulated with silicone grease so that only a short portion (50–100 µm long) of the fiber extremity remained out of the silicone. Fibers were bathed in a standard voltage-clamp extracellular solution containing (in mM) 140 TEA-methanesulfonate, 2.5 $CaCl_2$, 2 $MgCl_2$, 1 4-aminopyridine, 10 HEPES and 0.002 tetrodotoxin. An RK-400 patch-clamp amplifier (Bio-Logic, Claix) was used in whole-cell conFigureuration in combination with an analog-digital converter (Axon Instruments, Digidata 1440A) controlled by pClamp nine software (Axon Instruments). Voltage-clamp was performed with a micropipette filled with a solution containing (in mM) 120 K-glutamate, 5 $Na_2$-ATP, 5 $Na_2$-phosphocreatine, 5.5 $MgCl_2$, 15 EGTA, 6 $CaCl_2$, 0.1 rhod-2, five glucose, 5 HEPES. The tip of the micropipette was inserted through the silicone within the insulated part of the fiber and was gently crushed against the bottom of the chamber to ease intracellular equilibration and decrease the series resistance. Intracellular equilibration of the solution was allowed for 30 min before initiating measurements. Membrane depolarizing steps of 0.5 s duration were applied from −80 mV. Confocal imaging was conducted with a Zeiss LSM 5 Exciter microscope equipped with a 63x oil immersion objective (numerical aperture 1.4). Rhod-2 fluorescence was detected in line-scan mode (x,t, 1.15 ms per line) above 560 nm, upon excitation from the 543 nm line of a HeNe laser. Rhod-2 fluorescence transients were expressed as F/F0 where F0 is the baseline fluorescence. The Ca2+ release flux (rate of SR Ca2+ release) was estimated from the time derivative of the total myoplasmic Ca2+ ([Catot]) calculated from the occupancy of intracellular calcium binding sites following a previously described procedure (*Kutchukian et al., 2017*).

## q-PCR
### RNA extraction and RT-PCR
After the addition of Trizol (Sigma, T9424-200mL) on each sample, lysing matrix D and fast prep system (MPbio, 6913–100) were used for sample digestion and pre-RNA extraction. In order to extract RNA, samples were incubated in chloroform for 5 min at RT, centrifuged for 15 min at 12,000 rcf at 4°C and incubated in the tubes containing isopropanol (precipitatation of RNA) for 10 min at RT. following a centrifuge of samples for 15 min at 12,000 rcf at 4°C, samples were washed 2 times with 70% ethanol and the final RNA pellets were diluted in ultra-pure RNase free water (Invitrogen, 10977–035). RNA concentration was calculated using Nanodrop (ThermoFisher Scientifics). Goscript Reverse Transcriptase System (Promega, A5001) was used as described by the manufacturer to produce the cDNA.

### DNA extraction
DNA was extracted from frozen whole *Gastrocnimius* according to the manufacturers' protocol, using a Maxwell 16 Instrument (Promega Corporation, Madison, USA) and a Maxwell 16 Tissue DNA Purification Kit ((Promega Corporation, Madison, USA)). DNA concentration was calculated using Nanodrop (ThermoFisher Scientifics).

### q-PCR
Fast Start Universal SYBR Green Master (Rox)(Roche, 04913914001) and CFX Connect Real-Time PCR Detection System (BioRad) were used to carry out the quantitative PCR using the following primer sets. For analysis of genes expression level, The CT of target genes was normalized on four control genes. To compare the level of mitochondrial DNA, CT of mitochondrial DNA was normalized on genomic DNA.

List of the primares used for q-PCR:

| Primers | Sequence |
|---|---|
| MACF1 F | ATTGATTCACCGATACAGGCCC |
| MACF1 R | ATCTTCTGCATCTAGCAGTCGG |
| MyoD F | AGCACTACAGTGGCGACTCA |
| MyoD R | GCTCCACTATGCTGGACAGG |
| Myogenin F | CAATGCACTGGAGTTCGGTC |
| Myogenin R | ACAATCTCAGTTGGGCATGG |
| Sarcolipin F | GGTCCTTGGTAGCCTGAGTG |
| Sarcolipin R | CGGTGATGAGGACAACTGTG |
| DNM1 F | ATTTCGTGGGCAGGGACTTTC |
| DNM1 R | CAGTGCAGGAACTCGGCATA |
| DNM2 F | GGACCAGGCAGAGAATGAGG |
| DNM2 R | ACGTAGGAGTCCACCAGGTT |
| PPARGC1A F | GCAGGTCGAACGAAACTGAC |
| PPARGC1A R | CTTGCTCTTGGTGGAAGCAG |
| Gusb F | GAGGATTGCCAACGAAACCG |
| Gusb R | GTGTCTGGGGACCACCTTTGA |
| RpL4 F | GCCATGAGAGCGAAGTGG |
| RpL4 R | CTCCTGCAGGCGTCGTAG |
| Myoglobin F | CCTGGGTACTATCCTGAAGA |
| Myoglobin R | GAGCATCTGCTCCAAAGTCC |
| Betatub F (gDNA) | GCCAGAGTGGTGCAGGAAA |
| Betatub R (gDNA) | TCACCACGTCCAGGACAG |
| ND1 F (mDNA) | CCCAGCTACTACCATCATTCAAGT |
| ND1 R (mDNA) | GATGGTTTGGGAGATTGGTTGATGT |

### *Drosophila* model and sample preparation

Shot GD9507 UAS-RNAi line from VDRC collection crossed to Mef2-GAL4 driver has been used to attenuate shot gene expression specifically in muscles. Third instar larvae were dissected in physiological salt with 25 mM EDTA. Body wall muscles were fixed with 4% formaldehyde in PBS for 15 min and then rinsed three times for 5 min each in PBS with 0.5% Tween 20 (PBT). Muscles were blocked for 30 min with 20% horse serum in PBT at RT. Staining was performed by using primary antibodies applied overnight at 4°C and after washing three times in PBT secondary antibodies were applied at RT for 1 hr. The following primary antibodies were used: anti-Brp1 (1:100; DSHB, Nc82-s), anti-Shot (1:100; DSHB, mAbRod1).

### Electronic-microscopy

Tissues were cut into small pieces and fixed in 2% glutaraldehyde for 2 hr at 4°C. Samples were washed three times for 1H at 4°C and post-fixed with 2% $OsO_4$ 1 hr at 4°C. Then tissues were dehydrated with an increasing ethanol gradient (5 min in 30%, 50%, 70%, 95%) and tree times for 10 min in absolute ethanol. Impregnation was performed with Epon A (75%) plus Epon B (25%) plus DMP30 (1.7%). Inclusion was obtained by polymerization at 60°C for 72 hr. Ultrathin sections (approximately 70 nm thick) were cut on a UC7 (Leica) ultra-microtome, mounted on 200 mesh copper grids coated with 1:1000 polylysine, and stabilized for 1 day at room temperature and contrasted with uranyl acetate and lead citrate. Sections were acquired with a Jeol 1400JEM (Tokyo, Japan) transmission electron microscope, 80Kv, equipped with a Orius 600 camera and Digital Micrograph.

## Video-microscopy

To analyze the movement of myonuclei, time-lapse 10X images were acquired using Z1-AxioObserver (Zeiss) with intervals of 15 min. Final videos were analyzed using Metamorph (Zeiss) and Sky-Pad plugin as described before (*Cadot et al., 2014*).

To analyze the dynamic of EB3, 50 stream acquisitions were obtained at 63X with 500 ms of intervals using Z1-AxioObserver (Zeiss). The movement and speed of each comet was further analyzed as described before (*Sbalzarini and Koumoutsakos, 2005*).

## Quantification methods for myonuclei spreading in myotubes in vitro

Quantifications in immature myotubes were assessed using an analysis tool developed in our team. An image analysis performed in ImageJ software is combined with a statistical analysis in RStudio software. This provides quantifications of parameters, ranked by myonuclei content per myotubes, regarding phenotype of myotubes (area, length) and their respective myonuclei positioning compare to centroid of myotubes (DMcM).

MSG diagrams were obtained through the normalization of lengths of all analyzed myotubes (independently to their myonuclei content) to 100%. White lines represent myonuclei density curves assessing the statistical frequency for myonuclei positioning along myotubes. Each color group reflects statistical estimation of myonuclei clustering along myotubes.

## Quantification methods for fixed myofibers in vitro

Following fixation and immunofluorescent staining, the following steps were taken to generate data.

The measurement of distance between adjacent myonuclei, myonuclei feret and roundness were done on fibers stained for myonuclei and F-actin.

Microtubules intensity around the myonuclei was assessed on confocal images from total alpha tubulin. An ROI around each myonuclei was selected to measure the mean fluorescent intensity.

The width of each myofiber was measured on F-actin staining in myofibers presenting formed sarcomeres and the sarcomeric length was measured on alpha actinin staining.

Bungarotoxin staining was used to analyze formation of acetylcholine receptors clusters in mature myofibers. Representative images were taken from myofibers of each condition as described. Briefly, a threshold of 8 µm was set as the minimal size for AChRs clusters. Myofibers with at least one cluster of 8 µm or bigger were considered as positive.

Cytochrome C was used to study the mitochondrial content of each myofiber in vitro. On each myofiber, several Regions Of Interest (ROI) were selected randomly. Next, the total number of particles stained for Cytochrome C were quantified per ROI. The particles with circularity equivalent to one were eliminated in order to purify our results from any unwanted background errors. Particles with feret less than 0.75 µm (mean size of mitochondria) were eliminated to purify our results. Finally, the ratio of number of particles per area of ROI was calculated.

All the quantifications/measurements were done using ImageJ software (RRID:SCR_003070).

## Quantification methods for fixed myofibers isolated from control or conditional KO mice

After immunofluorescence staining, the distance between each myonuclei and its closest neighbor, myonuclei feret and myonuclei roundness were quantified on 3 µM peripheral Z stacks of each myofiber.

The microtubules orientation was assessed on mono Z images of tubulin staining using TeTD directionality tool as described before (*Liu and Ralston, 2014*).

To study the microtubules pool around myonuclei, 3 µM peripheral Z stacks of each myofiber were analyzed for tyrosinated or de-tyrosinated tubulin staining. Myonuclei presenting a complete de-tyrosinated tubulin ring around them were considered as positive for stable microtubule.

Each NMJs was studied on Z stack images of bungarotoxin staining as described previously (*Jones et al., 2016*). To study different pools of microtubules at the NMJs, Z series images were quantified for total, tyrosinated or de-tyrosinated alpha tubulin specifically at post-synaptic muscular region based on the bungarotoxin staining to select the NMJs ROI.

To analyze the organization of mitochondria and Desmin, 3 µM peripheral Z stacks of each myofiber were studied. Any fiber presenting broken or oriented organization instead of clear transversal/

longitudinal organization (that was observed in control myofibers) was considered as a myofiber with irregular staining.

All the quantifications/measurements were done using ImageJ software (RRID:SCR_003070).

## Statistical analysis

GraphPad Prism (RRID:SCR_002798) was used to perform statistical analysis (Student's t test or the Mann-Whitney test) according to samples amount and distribution.

# Acknowledgements

This work was supported by an ATIP-Avenir and MyoNeurAlp Alliance grants. We thank Pr. Ronald Liem and Dr. Alper (Columbia University of New York) for providing cKO *Macf1* embryos. We would like to thank Jijumon AS from the team of Dr. Carsten Janke (Institut Curie de Paris) for providing the MACF1-MTBD plasmid. We acknowledge the contributions of the CELPHEDIA Infrastructure (http://www.celphedia.eu/), especially the center AniRA of Lyon in addition to members of CIQLE imaging center (Faculté de Médecine Rockefeller, Lyon-Est). We thank Dr. Yoan Couté for the mass spectrometry analysis performed at EDyP Lab. We are grateful to Caroline Brun, Leonardo Beccari, Hélène Puccio and Bénédicte Chazaud for helping us with proofreading.

# Additional information

### Funding

| Funder | Grant reference number | Author |
|---|---|---|
| Programme AVENIR 2014 (ATIP-Avenir) | R14074CS | Vincent Gache |
| AFM-Téléthon | Alliance MyoNeurAlp | Vincent Gache |

The funders had no role in study design, data collection and interpretation, or the decision to submit the work for publication.

### Author contributions

Alireza Ghasemizadeh, Conceptualization, Formal analysis, Investigation, Visualization, Methodology, Writing - original draft, Writing - review and editing; Emilie Christin, Formal analysis, Visualization, Methodology; Alexandre Guiraud, Colline Sanchez, Vincent Jacquemond, Formal analysis, Investigation, Methodology; Nathalie Couturier, Francisco-Ignacio Jaque-Fernandez, Formal analysis, Methodology; Marie Abitbol, Formal analysis, Methodology, Writing - original draft; Valerie Risson, Laurent Schaeffer, Resources; Emmanuelle Girard, Resources, Methodology; Christophe Jagla, Cedric Soler, Lilia Laddada, Investigation; Jean-Luc Thomas, Marine Lanfranchi, Methodology; Julien Courchet, Resources, Methodology, Writing - original draft; Julien Gondin, Resources, Formal analysis, Investigation, Methodology; Vincent Gache, Conceptualization, Resources, Formal analysis, Supervision, Funding acquisition, Validation, Investigation, Visualization, Methodology, Writing - original draft, Project administration, Writing - review and editing

### Author ORCIDs

Alireza Ghasemizadeh (iD) https://orcid.org/0000-0002-8202-8546
Valerie Risson (iD) http://orcid.org/0000-0002-6812-6297
Emmanuelle Girard (iD) http://orcid.org/0000-0001-6730-9596
Cedric Soler (iD) http://orcid.org/0000-0002-2312-5370
Julien Courchet (iD) http://orcid.org/0000-0002-1199-9329
Vincent Gache (iD) https://orcid.org/0000-0002-2928-791X

### Ethics

Animal experimentation: All of the experiments and procedures were conducted in accordance with the guidelines of the local animal ethics committee of the University Claude Bernard - Lyon 1 and in

accordance with French and European legislation on animal experimentation and approved by the ethics committee CECCAPP (ref APAFIS#17455-2018091216033835v5) and the French Ministry of Research.

### Decision letter and Author response

Decision letter https://doi.org/10.7554/eLife.70490.sa1
Author response https://doi.org/10.7554/eLife.70490.sa2

## Additional files

### Supplementary files

• Transparent reporting form

### Data availability

All data generated or analysed during this study are included in the manuscript and supporting files. Source data files have been provided for all figures.

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
