## [Decision Letter]

**Acceptance summary:**

This paper provides important data on the role of skeletal muscle MACF1 (a protein related to the microtubule network) in the positioning of myonuclei, the structural integrity of the mitochondria, the clustering of AChR, and the stability of the neuromuscular junction. These results are broadly significant for the field of muscle physiology and the biology of the neuromuscular system. The article will be of interest both to muscle physiologists and to neuroscientists studying synapses.

**Decision letter after peer review:**

[Editors’ note: the authors submitted for reconsideration following the decision after peer review. What follows is the decision letter after the first round of review.]

Thank you for submitting your work entitled "Muscle MACF1 maintains myonuclei and mitochondria localization through microtubules to control muscle functionalities" for consideration by *eLife*. Your article has been reviewed by 3 peer reviewers, and the evaluation has been overseen by a Reviewing Editor and a Senior Editor. The reviewers have opted to remain anonymous.

Our decision has been reached after consultation between the reviewers. Based on these discussions and the individual reviews below, we regret to inform you that your work will not be considered for publication in *eLife* in its present form. The maximum time for revision is two months and we feel that you will require more time to answer the questions posed by the reviewers. We therefore encourage you to resubmit the revised manuscript. We think that the suggestion that MACF1 is involved in microtubule dynamics is novel and interesting. This aspect should be developed. In contrast MACF1 implication in myonuclear positioning has already been proposed. Its role in mitochondrial localisation would require consolidating if you want to make this a major point of the paper. We appreciate all the in vitro and in vivo analyses presented in the paper. In this context, is not clear to us why knock-down in vitro produces such a rapid effect whereas in vivo it takes months to see an effect. Is this linked first to NMJ defects due to the role of MACF1 in microtubule dynamics? In addition to addressing this question, you should provide more information on the subcellular localisation of MACF1and its relative association with microtubules, actin fibres and the nuclear membrane. The reports provided by the three reviewers are shown below. In the revised paper you should consider the different points raised and indicate how you have addressed them in an accompanying letter.

*Reviewer #1:*

In this manuscript, Ghasemizadeh et al. investigated the role of the skeletal muscle MACF1, a protein related to the microtubule network, in the maintenance of myonuclei and mitochondria localization in mature muscle cells. The major findings are: (1) MACF1 regulates the myonuclei positioning during maturation steps of myofibers but not during their early steps of formation, (2) MACF1 controls the localization and structural integrity of muscle mitochondria, (3) the knockdown of MACF1 in myotubes with shRNA reduces , the formation of agrin-induced AChR clusters and increases myonuclei movements, (4) the loss-function of MACF1 specifically in muscles (conditional muscle-KO) induces mis-localization of myonuclei in adult but not in younger mice, alters the structural integrity of NMJs at younger mice, and enhances mitochondrial content in in adult mice, (7) muscles from Macf1 conditional mice display a higher resistance to fatigue compared to muscles of control mice.

Overall, this is an interesting study addressing an important question and provides important mechanistic insight into the role of MACF1 muscle in the positioning of myonuclei and the structural integrity of the mitochondria and the neuromuscular junction.

The presented data appears to be compelling and is well discussed in light of previous literature. However, some issues need to be addressed to enhance the visibility of the manuscript.

1. It would be helpful to know about the subcellular location of MACF1 in muscle cells (if the antibody does work in immunocytochemistry).

2. In Figure1 D and E, MHC-positive cells appear to be at low frequency, indicating that most of cells remain as non-differentiated myoblats or dead cells. This raises the question of whether myoblats or myotubes are the primary source of MACF1 protein expression observed in total protein extracts by Western blot. The authors should resolve this issue.

3. The authors claimed that MACF1 is progressively expressed and accumulates during muscle differentiation and maturation processes. However, there is no provided data to support this claim. On the contrary, MACF1 expression level appears to be less pronounced at 10 days of muscle differentiation (Figure 1 C). It would useful to examine MACF1 expression levels during muscle cell differentiation and maturation processes both in vivo and in vitro.

4. In Figure 2, the authors showed that the number of AChR dots was significantly increased in myotubes treated with MAFC1 shRNA. However, the unusual shape and size of fluorescent BTX raised some concerns about the specificity of the staining. This can be ruled out by saturating all AChRs with unlabeled BTX and then add the fluorescently tagged BTX. In this case, you should expect to see no staining.

5. It was puzzling to see that mislocalization of myonuclei in MACF1 conditional muscle-KO occurs only after a year, while at the NMJ the number of positioned subsynaptic nuclei was altered much earlier (4 months). The authors should explore the idea whether the positioning of non-synaptic and subsynaptic myonuclei is MCF1 activity dependent. One possible experiment is to denervate muscles of mutant and wild type and see the positioning/number of myonuclei.

6. The authors should also look if the synaptic mitochondria are susceptible to alterations in younger MACF1 conditional muscle-KO mice.

*Reviewer #2:*

In this manuscript Ghasemizadeh et al., analyzed the function of MACF1, a microtubule-actin crosslinker protein in muscles. The authors used shRNA knock down in culture conditions, as well as muscle specific knock out in a mouse model for their analysis. They show that myonuclei positioning in vitro and in vivo is impaired in the conditional knock out mouse muscles, and that mitochondria positioning and morphology was impaired. Both phenotypes might be related to destabilized dynamics of the microtubules in the muscles as observed by the authors. In addition abnormal acetylcholine receptors clustering at the neuromuscular junction was observed.

The results described are interesting, however they might be unrelated to each other, and their novelty is limited. The authors do not provide novel mechanistic insight to any of the processes described, but rather provide a collection of observations and phenotypes observed following knockout of MACF1 in muscles.

1. The authors used MACF1 conditional knockout mice described in Goryunov et al., 2010. In this model the authors deleted exons 6 and 7 encoding the C-terminal half of the MACF1 actin binding domain (ABD). MACF1 is a large protein with distinct domains, including ABD, Plakin, microtubule and EB1 domains. The authors suggest that the major contribution of MACF-1 in the muscles relates to its function as a microtubule binding protein. However, the truncated protein (produced in the KO mice), lacks only part of the ABD domain, and still contains functional plakin and microtubule binding domains. It is possible that the weak phenotype observed is under representative of the major function of MACF1 in muscles. The authors did not address or discuss this possibility.

2. Many of the phenotypes described by the authors have been already illustrated in other model organisms, including myonuclear positioning defects, myonuclear sphericity, and microtubule disorganization (e.g. Wang et al., 2015; Kim et al., 2010, Bottenberg et al., 2009). The phenotype of the mitochondria fragmentation is novel and interesting, however, the authors do not provide mechanistic explanation for this observation, and only speculate that this is due to aberrant microtubule organization.

3. Some of the images are not convincing: Figure 1 – the difference between control and MACF1 siRNA are minimal and might be due to differential length of the myofibers. Figure 2 – actin does not appear normal. The difference in microtubule organization is not clear enough. The EM micrographs are of medium quality, and do not allow the reader to see the details of the mitochondria. Higher magnification images are required.

4. The authors should show where MACF1 is localized in a normal muscle. Does it overlaps with MT? Is it localized at the nuclear membrane`? does it overlaps with tyrosinated or with Non-tyrosinated? Does it overlap with EB1?

*Reviewer #3:*

With the exception of the clustered myonuclei at the neuromuscular junction (NMJ), multinucleated skeletal muscle cells position most of their nuclei to optimize the distances between them. Moreover, the skeletal muscle cells have developed mechanisms to maintain these positions despite muscle contraction. The manuscript focuses on identifying the role of the microtubule binding protein, MACF1 in mouse muscle cells through a series of in vitro and in vivo experiments. The investigators report that MACF1 is linked to myonuclear positioning at the NMJ and throughout the fiber, mitochondrial positioning/structure, T-tubule organization, and the clustering of Acetylcholine receptors at the NMJ. The authors suggest that the key role of MACF1 is to regulate posttranslational modifications of microtubules and, as a result, microtubule dynamics. They make the argument that alterations in microtubule dynamics found in the loss of MACF1 are linked to deficits in postsynaptic nuclear positioning and function, myonuclear and mitochondrial positioning and ultimately muscle function.

The authors provide a great deal of in vivo and in vitro data to develop their assertion that MACF1 plays a critical role in muscle maturation and function. They find that loss/reduction of MACF1 results in a number of deficits in the mature fiber: these include morphological defects including aberrant myonuclear position, particularly of the postsynaptic nuclei, reduction in ACHR clustering at the synapse, mitochondria mispositioning and morphology, T-tubule abnormalities, etc and muscle functional defects (reduced force but greater endurance). The authors attempt to link myriad defects in the muscle fiber to one underlying structural alteration in the fibers: abnormal microtubule dynamics. Their evidence to support this: changes in the amount of tyrosination/detyrosinated microtubules (which should be quantified but isn't) and alterations in EB1 comet speed (number and trajectory numbers should be included as well). Both these are suggestive of highly dynamic microtubules in the ko/kd myofibers and some role for MACF1 in the control of microtubule dynamicity (either directly or indirectly). Given that the main conclusions rest on the conclusion of MACF1 control microtubule dynamics, some additional support should be provided that connects MACF1 to Microtubules to Microtubule dynamics and subsequently these altered MTs to the other phenotypes described. For example, does overexpression of MACF1 lead to an increase in stable microtubules? How does this affect AChR clustering, myonuclear clustering at the synapse or even T-tubule structures? Does MACF1 recruit enzymes that post translationally modify MTs to the MTs? Where is MACF1 expressed or enriched in the mouse myofiber at the different ages?

Also important: please have a native English speaker edit the manuscript as there are many issues in grammar, word choice, sentence structure. See also some of the comments listed below for each figure.

---

## [Author Response]

[Editors’ note: the authors resubmitted a revised version of the paper for consideration. What follows is the authors’ response to the first round of review.]

Reviewer #1:In this manuscript, Ghasemizadeh et al. investigated the role of the skeletal muscle MACF1, a protein related to the microtubule network, in the maintenance of myonuclei and mitochondria localization in mature muscle cells. The major findings are: (1) MACF1 regulates the myonuclei positioning during maturation steps of myofibers but not during their early steps of formation, (2) MACF1 controls the localization and structural integrity of muscle mitochondria, (3) the knockdown of MACF1 in myotubes with shRNA reduces , the formation of agrin-induced AChR clusters and increases myonuclei movements, (4) the loss-function of MACF1 specifically in muscles (conditional muscle-KO) induces mis-localization of myonuclei in adult but not in younger mice, alters the structural integrity of NMJs at younger mice, and enhances mitochondrial content in in adult mice, (7) muscles from Macf1 conditional mice display a higher resistance to fatigue compared to muscles of control mice.Overall, this is an interesting study addressing an important question and provides important mechanistic insight into the role of MACF1 muscle in the positioning of myonuclei and the structural integrity of the mitochondria and the neuromuscular junction.The presented data appears to be compelling and is well discussed in light of previous literature. However, some issues need to be addressed to enhance the visibility of the manuscript.

We thank the reviewer for his positive comments and hope that all his concerns are now addressed.

1. It would be helpful to know about the subcellular location of MACF1 in muscle cells (if the antibody does work in immunocytochemistry).

To define the subcellular localization of MACF1, we performed immunostaining on muscle samples (in vitro and ex vivo myofibers) using the rabbit polyclonal antibody from Novus biologicals.

As presented in Figure 1B, the immunostaining revealed that MACF1 is concentrated at the periphery of in vitro mature primary myofibers. As well, it appears as dots around the myonuclei, following lines that colocalize with the microtubule network (mostly at the cortex of myofibers and also inside myofibers). Interestingly, no colocalization of MACF1 is observed with the actin network. Confirming the specificity of the antibody, MACF1 immunostaining was strongly decreased in *Macf1*-shRNA treated myofibers (Figure 1G).

In agreement with these data, ex vivo isolated *Tibialis Anterior* myofibers from wild-type mice showed a clear perinuclear accumulation of MACF1 (Figure 3D, upper panels). In addition, MACF1 colocalizes with both the longitudinal and transversal microtubule network but not with the actin network (Figure 3D, upper panels). As expected, MACF1 labeling was almost entirely lost in the muscle-specific MACF1 conditional KO mouse model (Figure 3D, lower panels).

Thanks to this immunostaining approach, we have now been able to point out the subcellular localization of MACF1, *i.e.* around myonuclei and in association with microtubules in skeletal muscle fibers.

2. In Figure1 D and E, MHC-positive cells appear to be at low frequency, indicating that most of cells remain as non-differentiated myoblats or dead cells. This raises the question of whether myoblats or myotubes are the primary source of MACF1 protein expression observed in total protein extracts by Western blot. The authors should resolve this issue.

To address the question, we analyzed the presence of MACF1 protein in primary muscle cells as they differentiate into myotubes (Figure 1C). In growing (and undifferentiated) myoblasts (GM), no MACF1 was detected. However, we clearly observed the presence of MACF1 after 3-days and 5-days of differentiation (Figure 1C). Thus, these results show that myotubes are the primary source of MACF1 during the differentiation process. This panel has now been transferred in supplementary Figure 1E.

3. The authors claimed that MACF1 is progressively expressed and accumulates during muscle differentiation and maturation processes. However, there is no provided data to support this claim. On the contrary, MACF1 expression level appears to be less pronounced at 10 days of muscle differentiation (Figure 1 C). It would useful to examine MACF1 expression levels during muscle cell differentiation and maturation processes both in vivo and in vitro.

To examine MACF1 expression levels in vitro, we performed western blotting on both primary muscle cells (Figure 1C and supplementary Figure 1C) and C2C12 myogenic cell line (supplementary Figure 1D). Clearly, we observed a burst in MACF1 expression when comparing proliferative myoblasts and 3-days differentiated myotubes (Figure 1C and supplementary Figure 1C, 1D). However, MACF1 protein levels was decreased in primary cells from 3 to 5 days of differentiation. in vivo, we conducted a qPCR approach on samples from *Tibialis Anterior* muscles of control (Cre-) and *Macf1*-cKO (Cre+) mice, confirming that *Macf1* mRNA expression is relatively stable in mature muscle (Author response image 1).

Given these observations, the manuscript has been corrected, highlighting the burst in MACF1 expression as muscle cells differentiate (lines 120-123).

**Author response image 1. sa2fig1:** QRT-PCR results presenting relative gene expression level (to housekeeping genes) for MACF1 in *Macf1cKO* Cre- and Cre+ at each given age.

4. In Figure 2, the authors showed that the number of AChR dots was significantly increased in myotubes treated with MAFC1 shRNA. However, the unusual shape and size of fluorescent BTX raised some concerns about the specificity of the staining. This can be ruled out by saturating all AChRs with unlabeled BTX and then add the fluorescently tagged BTX. In this case, you should expect to see no staining.

Α-Bungarotoxin (aBTX) is a snake venom that targets specifically Acetylcholine Receptor (AchR). When aBTX is applied on ex vivo extracted mature myofibers, it binds and stains specifically the post-synaptic zone of NMJs (Tzartos SJ & Changeux JP, 1983, PMID: 11894953). In Figure 2, aBTX was directly applied on mature myofibers. In this condition, aBTX staining exhibit long (more than 10 µm) or dotty AChR clusters (Vilmont *et al.,* 2016, PMID: 27283349). To further confirm the specificity of our staining, in vitro developed mature myofibers treated with scramble or *Macf1* shRNA were first labeled with BTX coupled to Alexa-flour 555 (BTX-AF555, dilution 1/500), washed, and then labeled with BTX coupled to Alexa-flour 647 (BTX-AF647, dilution 1/500).

As presented in Author response image 2, a strong signal was detected for aBTX-AF555, whereas almost no signal was visible with aBTX-AF647. This confirmed the reliability of our aBTX staining.

**Author response image 2. sa2fig2:** aBTX staining in primary myofibers transfected with scramble or MACF1 shRNAs. Scale Bar 10 µm.

5. It was puzzling to see that mislocalization of myonuclei in MACF1 conditional muscle-KO occurs only after a year, while at the NMJ the number of positioned subsynaptic nuclei was altered much earlier (4 months). The authors should explore the idea whether the positioning of non-synaptic and subsynaptic myonuclei is MCF1 activity dependent. One possible experiment is to denervate muscles of mutant and wild type and see the positioning/number of myonuclei.

We thank the reviewer for this comment that allowed us to better discriminate which myonuclei are under the control of MACF1 in myofibers.

Our previous data showed that myonuclei were internalized in 12-months old mice. To decipher if extra-synaptic myonuclei localization was affected, we first extracted *Tibialis Anterior* muscle fibers from *MACF1* cKO mouse and quantified the distance between myonuclei in 2, 6 and 12months old mice. This approach showed that extra-synaptic myonuclei positioning is affected in *MACF1* cKO mouse compared to controls. Indeed, we observed a 20% reduction in the distance between closest myonuclei at 2 months, 40% at 6 months and up to 70% in 12 months of age (Figure 4B). Thus, as suggested by our in vitro results (Figure 1J), alteration of extra-synaptic myonuclei spreading at the periphery of myofiber occurs precociously. In contrast, internalization of myonuclei is a late event, concomitant with microtubule and intermediate filaments disorganization (Figure 3 and 7G).

In a second approach, we quantified the number of synaptic myonuclei (beneath the NMJ) during aging. Interestingly, we found that the number of synaptic myonuclei is increased in control mice, from 7 in young mice to 11 in old mice, while this number is relatively stable (around 7) in *MACF1* cKO mice (Figure 5D). As NMJs in *MACF1* KO mice are smaller and present fragmented AChRs clusters compared to control (Figure 5B-C), we hypothesized that failure in synaptic myonuclei recruitment observed in *MACF1* cKO mouse is preferentially driven by the failure in AChR Clustering (Figure 5B-C). To validate this hypothesis, we performed a muscle denervation experiment. The sciatic nerve crush, as previously published (Milanič et al., 1999, PMID: 10421467 ), decreases both myofiber (Cross Section Area, Figure 5H) and NMJs area (Figure 5I) after 15 days. These effects were similar in wild-type and in *MACF1* cKO mouse (Figure 5H-I). We also observed a similar loss of synaptic myonuclei at the NMJ after the denervation experiment (1 synaptic myonuclei), suggesting that MACF1 is not directly involved in synaptic myonuclei positioning at the NMJs.

Therefore, these data show that MACF1 regulates directly the dynamics of extra-synaptic myonuclei at the periphery of muscle fibers rather than controlling the synaptic myonuclei.

6. The authors should also look if the synaptic mitochondria are susceptible to alterations in younger MACF1 conditional muscle-KO mice.

To address this question, we investigated the organization of mitochondria on isolated myofibers extracted from wild-type and *MACF1* cKO old mice. As presented in Figure 7 G and H, the overall organization of mitochondria is altered in adult *MACF1* cKO mice compared to controls. However, we did not observe changes in the density of mitochondria at the NMJs in mice lacking MACF1 compared to control mice (Author response image 3).

Therefore, these data show that MACF1 alters the organization of non-synaptic mitochondria.

**Author response image 3. sa2fig3:** Representative confocal microscopy images (63X) from mitochondrial density (reflected by Cytochrome-C staining) at NMJs (reflected by BTX staining) of extracted *Tibialis anterior* of 12-month-old MACF1 Creand Cre+ mice. In the overlay, mitochondria are presented in green, postsynaptic NMJs in red and synaptic nuclei in blue. Scale Bar 10 µm.

Reviewer #2:In this manuscript Ghasemizadeh et al., analyzed the function of MACF1, a microtubule-actin crosslinker protein in muscles. The authors used shRNA knock down in culture conditions, as well as muscle specific knock out in a mouse model for their analysis. They show that myonuclei positioning in vitro and in vivo is impaired in the conditional knock out mouse muscles, and that mitochondria positioning and morphology was impaired. Both phenotypes might be related to destabilized dynamics of the microtubules in the muscles as observed by the authors. In addition abnormal acetylcholine receptors clustering at the neuromuscular junction was observed.The results described are interesting, however they might be unrelated to each other, and their novelty is limited. The authors do not provide novel mechanistic insight to any of the processes described, but rather provide a collection of observations and phenotypes observed following knockout of MACF1 in muscles.

We respectfully disagree with this reviewer as our work presents the hitherto unexplored role of MACF1 to control microtubule dynamics/stability and consequently, extra-synaptic myonuclei positioning, AchRs clustering and mitochondria biogenesis. We have now provided new mechanistic data that further support the essential role of MACF1 for maintaining myonuclei localization and overall skeletal muscle fiber function.

1. The authors used MACF1 conditional knockout mice described in Goryunov et al., 2010. In this model the authors deleted exons 6 and 7 encoding the C-terminal half of the MACF1 actin binding domain (ABD). MACF1 is a large protein with distinct domains, including ABD, Plakin, microtubule and EB1 domains. The authors suggest that the major contribution of MACF-1 in the muscles relates to its function as a microtubule binding protein. However, the truncated protein (produced in the KO mice), lacks only part of the ABD domain, and still contains functional plakin and microtubule binding domains. It is possible that the weak phenotype observed is under representative of the major function of MACF1 in muscles. The authors did not address or discuss this possibility.

Indeed, Goryunov *et al.* observed that conditional deletion of the exons 6 and 7 of *Macf1* does not eliminate the presence of the MACF1c isoform. To our knowledge, this isoform is unique to the nervous system (Cusseddu *et al.*, 2021, PMID: 33816492). Supporting this data, we did not observe the presence of such isoform or truncated protein in our muscle system (Figure 1c and supplementary Figure 1c). Therefore, the MACF1c isoform cannot be responsible of the mild phenotype observed.

In muscle-specific MACF1 conditional KO mouse, qPCR analysis demonstrated that *Macf1* mRNA expression was reduced by 50% in 2-month-old mice and 70% in 12-month-old mice (supplementary Figure 3A). The partial reduction in *Macf1* expression might be attributable to several parameters. First, qPCR experiments were performed on total muscle extracts, which do not reflect solely *Macf1* expression in myofiber. Indeed, skeletal muscle contains multiple cell types in which *Macf1* expression is not altered in muscle-specific MACF1 conditional KO mouse. Then, muscle stem cells contribute to uninjured myofiber homeostasis throughout life (Keefe *et al.*, 2015, PMID: 25971691, Pawlikowski *et al.*, 2015, PMID: 26668715), thus providing new myonuclei where MACF1 is not depleted yet. As well, *Macf1* mRNA could be brought in the mature myofiber upon muscle stem cell fusion. Together, this could partly dilute the effect of the Cre recombinase and allow maintenance of small amounts of MACF1 protein in myofibers.

2. Many of the phenotypes described by the authors have been already illustrated in other model organisms, including myonuclear positioning defects, myonuclear sphericity, and microtubule disorganization (e.g. Wang et al., 2015; Kim et al., 2010, Bottenberg et al., 2009). The phenotype of the mitochondria fragmentation is novel and interesting, however, the authors do not provide mechanistic explanation for this observation, and only speculate that this is due to aberrant microtubule organization.

As presented in the first version of our manuscript, we observed an increase in the number of mitochondrial particles following MACF1 downregulation. This phenomenon, which was referred as “mitochondrial fragmentation”, led us to hypothesize that loss of MACF1 disrupts the mitochondrial network integrity. By disorganizing the microtubule (MT) network, MACF1 downregulation would affect the expression or activity level of MT-associated proteins such as DNM1 or DNM2, known to participate in mitochondrial fission or fusion.

To investigate this hypothesis, we carried out different experiments as follows. First, the mRNA expression of mitochondrial fission/fusion actors were studied both in vitro and in vivo. No differences were observed in *Dnm1* or *Dnm2* expression levels when MACF1 is downregulated either in vitro (Author response image 4) or in vivo in the *MACF1* cKO mouse (Figure 7F). Therefore, loss of MACF1 does not affect mitochondrial fission/fusion processes.

**Author response image 4. sa2fig4:** QRT-PCR results presenting relative gene expression level of DNM1 (left) and DNM2 (right) to housekeeping genes in short differentiated myotubes or long term differentiated myofibers transfected with scrambled si/shRNA or a pool of si/shRNAs against MACF1.

However, the in vivo histo-enzymological SDH staining (Figure 7B) and q-PCR analysis of mitochondrial DNA amount (Figure 7E) revealed an increase in the number of mitochondria in the adult MACF1 KO mice compared to the control mice. This suggests a potential increase of mitochondrial biogenesis.

Interestingly, different studies have linked the increase of Sarcolipin (a Sarcoplasmic reticulum associated protein in skeletal muscle) to mitochondrial biogenesis (Santosh K. Maurya et al., 2015, PMID: 25713078; Santosh K. Maurya et al., 2018, PMID: 30208317). When we assessed the expression level of *Sarcolipin* in the *MACF1* cKO mouse model, we observed that *Sarcolipin* mRNA exhibit a 40-fold increase in adult MACF1 KO mice muscle compared to control (Figure 7F).

Altogether, these data show that loss of MACF1 increases mitochondrial biogenesis rather than modifying mitochondrial fusion, fission or fragmentation. We clarified our conclusion and revised the text accordingly. The increase in the number of mitochondrial particles in vitro is now introduced as an increase in mitochondrial particles content.

3. Some of the images are not convincing: Figure 1 – the difference between control and MACF1 siRNA are minimal and might be due to differential length of the myofibers. Figure 2 – actin does not appear normal. The difference in microtubule organization is not clear enough. The EM micrographs are of medium quality, and do not allow the reader to see the details of the mitochondria. Higher magnification images are required.

We agree with the reviewer and have now provided high magnification images that strengthen our conclusions.

For the reduced distance of adjacent myonuclei in MACF1-KD myofibers compared to control myofibers (Figure 1J), there is no apparent difference in myofiber length and/or shape. We added a larger view of the cell culture to better appreciate unchanged culture of myofibers (Figure 1D), To investigate actin organization in skeletal muscle fibers, the sarcomere length was measured on images using F-actin staining in control and MACF1-KD myofibers (Figure 1F). No significant difference was observed, showing that MACF1 is not involved in actin network patterning in myofibers. These results are coherent with our in vivo results presenting equal sarcomere length in wild-type and in *MACF1* cKO mice (Figure 3E and supplementary Figure 3F).

To analyze changes in the microtubule network around myonuclei, the intensity of microtubules was measured at the perinuclear region. As presented in Figure 1H, absence of MACF1 reduces microtubule intensity around myonuclei up to 50%. Panels of Figure 1G have been added to visually highlight the difference.

For electronic microscopy, images with higher resolution present the accumulation of round shaped mitochondria between myofibrils in the MACF1 KO mice compared to the control mice (Figure 7J-K).

4. The authors should show where MACF1 is localized in a normal muscle. Does it overlaps with MT? Is it localized at the nuclear membrane`? does it overlaps with tyrosinated or with Non-tyrosinated? Does it overlap with EB1?

Please refer to the comment #1 reviewer #1.

Immunostaining was performed to define MACF1 localization in myofibers in vitro and in vivo. in vitro, MACF1 is concentrated around myonuclei and colocalizes with cortical microtubules in mature primary myofibers in vitro (Figure 1B). In addition, we showed that the Microtubule Binding Domain of MACF1 tagged with GFP colocalizes with tyrosinated tubulin in immature myotubes in an end-binding protein comet-like pattern (Figure 2A).

in vivo, immunostaining revealed a clear accumulation of MACF1 around myonuclei in isolated myofibers of *Tibialis Anterior* of WT mice (Figure 3D, upper panels). As well, MACF1 colocalizes with both longitudinal and transversal microtubules in addition to perinuclear region. These results point out a specific localization for MACF1 around myonuclei and in association with microtubules in skeletal muscle fibers.

Reviewer #3:With the exception of the clustered myonuclei at the neuromuscular junction (NMJ), multinucleated skeletal muscle cells position most of their nuclei to optimize the distances between them. Moreover, the skeletal muscle cells have developed mechanisms to maintain these positions despite muscle contraction. The manuscript focuses on identifying the role of the microtubule binding protein, MACF1 in mouse muscle cells through a series of in vitro and in vivo experiments. The investigators report that MACF1 is linked to myonuclear positioning at the NMJ and throughout the fiber, mitochondrial positioning/structure, T-tubule organization, and the clustering of Acetylcholine receptors at the NMJ. The authors suggest that the key role of MACF1 is to regulate posttranslational modifications of microtubules and, as a result, microtubule dynamics. They make the argument that alterations in microtubule dynamics found in the loss of MACF1 are linked to deficits in postsynaptic nuclear positioning and function, myonuclear and mitochondrial positioning and ultimately muscle function.The authors provide a great deal of in vivo and in vitro data to develop their assertion that MACF1 plays a critical role in muscle maturation and function.

We thank the reviewer for appreciating the amount of our data provided to support our conclusion.

They find that loss/reduction of MACF1 results in a number of deficits in the mature fiber: these include morphological defects including aberrant myonuclear position, particularly of the postsynaptic nuclei, reduction in ACHR clustering at the synapse, mitochondria mispositioning and morphology, T-tubule abnormalities, etc and muscle functional defects (reduced force but greater endurance). The authors attempt to link myriad defects in the muscle fiber to one underlying structural alteration in the fibers: abnormal microtubule dynamics. Their evidence to support this: changes in the amount of tyrosination/detyrosinated microtubules (which should be quantified but isn't) and alterations in EB1 comet speed (number and trajectory numbers should be included as well). Both these are suggestive of highly dynamic microtubules in the ko/kd myofibers and some role for MACF1 in the control of microtubule dynamicity (either directly or indirectly). Given that the main conclusions rest on the conclusion of MACF1 control microtubule dynamics, some additional support should be provided that connects MACF1 to Microtubules to Microtubule dynamics and subsequently these altered MTs to the other phenotypes described. For example, does overexpression of MACF1 lead to an increase in stable microtubules? How does this affect AChR clustering, myonuclear clustering at the synapse or even T-tubule structures? Does MACF1 recruit enzymes that post translationally modify MTs to the MTs? Where is MACF1 expressed or enriched in the mouse myofiber at the different ages?

We have performed a number of additional experiments to strengthen our conclusion that MACF1 controls microtubule dynamics and its loss inevitably leads to functional deficits of the mature myofiber.

For the localization of MACF1 in muscle fibers, similar questions were raised by reviewer #1/question #1 and reviewer #2/question #4. We thank all of our three reviewers to have pointed out the missing of such an important information. Thus, the use of a rabbit polyclonal antibody from Novus biologicals, also validated for its specificity by western blot, allowed us to address MACF1 localization on in vitro and on ex vivo muscle fibers (Figures 1 and 3). Using this immunostaining approach, we validated MACF1 as an exclusive Microtubule-BindingProtein that colocalizes with the microtubule network and accumulates preferentially at the perinuclear area. As MACF1 is a huge protein with the main two isoforms at 830 and 630kD, in vitro over-expression of the full length MACF1 using a plasmid is experimentally impossible. However, we overexpressed the microtubule binding domain (MTBD) of MACF1 in our in vitro system. MACF1-MTBD formed bundles with a preference location around myonuclei (Figure 2B). This indicates that MACF1-MTBD is responsible for the perinuclear positioning of endogenous MACF1.

Furthermore, we analyzed EB3 comets speed along myofibers maturation in control and in MACF1 depleted myofibers (Figure 2C). These data show (1) that during myofibers formation, EB3 comet speed decreases and (2) that MACF1 depletion maintains EB3 comets speed to high level. All together, these data demonstrate that MACF1 control microtubule dynamic rate along myofiber maturation.

Also important: please have a native English speaker edit the manuscript as there are many issues in grammar, word choice, sentence structure. See also some of the comments listed below for each figure.

Our manuscript has now been proofread by all our collaborators.